# Noradrenergic consolidation of social recognition memory is mediated by β-arrestin–biased signaling in the mouse prefrontal cortex

Deqin Cheng[1,2], Junwen Wu[1,2], Enhui Yan[1,2], Xiaocen Fan[1,2], Feifei Wang[1,2], Lan Ma ![ORCID][1,2 ✉] & Xing Liu ![ORCID][1,2 ✉]

Social recognition memory (SRM) is critical for maintaining social relationships and increasing the survival rate. The medial prefrontal cortex (mPFC) is an important brain area associated with SRM storage. Norepinephrine (NE) release regulates mPFC neuronal intrinsic excitability and excitatory synaptic transmission, however, the roles of NE signaling in the circuitry of the locus coeruleus (LC) pathway to the mPFC during SRM storage are unknown. Here we found that LC-mPFC NE projections bidirectionally regulated SRM consolidation. Propranolol infusion and β-adrenergic receptors (β-ARs) or β-arrestin2 knockout in the mPFC disrupted SRM consolidation. When carvedilol, a β-blocker that can mildly activate β-arrestin-biased signaling, was injected, the mice showed no significant suppression of SRM consolidation. The impaired SRM consolidation caused by β1-AR or β-arrestin2 knockout in the mPFC was not rescued by activating LC-mPFC NE projections; however, the impaired SRM by inhibition of LC-mPFC NE projections or β1-AR knockout in the mPFC was restored by activating the β-arrestin signaling pathway in the mPFC. Furthermore, the activation of β-arrestin signaling improved SRM consolidation in aged mice. Our study suggests that LC-mPFC NE projections regulate SRM consolidation through β-arrestin-biased β-AR signaling.

[1] School of Basic Medical Sciences, State Key Laboratory of Medical Neurobiology, MOE Frontiers Center for Brain Science, Institutes of Brain Science, Department of Neurology, Pharmacology Research Center, Huashan Hospital, Fudan University, 200032 Shanghai, China. [2] Research Unit of Addiction Memory, Chinese Academy of Medical Sciences (2021RU009), 200032 Shanghai, China. ✉email: lanma@fudan.edu.cn; xingliu@fudan.edu.cn

Social recognition memory (SRM) is critical for establishing and maintaining social relationships[1], which are essential for environmental adaptation, reproduction, and survival[2,3]. In rodents, SRM is assessed by the ability to distinguish and prefer novel conspecifics over familiar conspecifics that have previously been encountered. As with other forms of learning, social information is acquired and consolidated when labile traces are stabilized into long-term memory[4]. The mechanisms underlying SRM consolidation are unclear.

The medial prefrontal cortex (mPFC) is critical for a broad range of social behaviors[1,5]. The mPFC is included in several neurocircuits that are crucial for social memory, including the ventral hippocampus–mPFC, amygdala–mPFC, and olfactory–mPFC circuits[6–8]. Infusions of NMDA- or AMPA/kainate-receptor antagonists into the mPFC prevent SRM consolidation[9]. This evidence suggests that the glutamatergic synaptic transmission in the mPFC regulates social ability and social recognition memory. Neuronal excitability in the mPFC is also critical for social memory. The excitatory neurons in the mPFC form distinct ensembles that are tuned toward social exploration and convey information about social targets[10]. Activation of excitatory neurons in the mPFC decreases social exploration[11]. Chemogenetic activation of mPFC excitatory neurons in Shank3-deficient mice restores reduced social recognition[12]. Protein synthesis in the mPFC is required for SRM consolidation, but not social recognition[13]. Norepinephrine (NE) is a common neuromodulator that plays a key role in attention, perception, and cognition[14]. The noradrenaline neurons in the locus coeruleus (LC), the main source of NE in the central nervous system, broadly project to the forebrain, including the mPFC through highly ramified axonal arborization[15,16]. Patch-clamp recordings have shown that NE release enhances mPFC neuronal intrinsic excitability[17], which is blockaded by β-adrenergic receptor (β-AR) antagonist[18]. β-AR activation also facilitates the postsynaptic responses of excitatory synapses on pyramidal neurons in the mPFC[19]. However, whether the LC-mPFC NE system is involved in SRM and how NE signaling in the mPFC could contribute to the SRM consolidation remain unknown.

β-ARs are prototypical heterotrimeric guanine nucleotide-binding protein (G protein)-coupled receptors (GPCRs) that respond to NE. Ligand binding induces GPCR conformational changes that recruit heterotrimeric G proteins, leading to the activation of adenylate cyclase. Furthermore, ligand-dependent phosphorylation of GPCR promotes β-arrestin recruitment, which induces receptor desensitization[20]. Moreover, β-arrestin can act as a scaffold protein, initiating signaling pathways independent of G proteins[21]. Our previous studies showed that β-AR/β-arrestin-biased signaling in the entorhinal cortex mediates the reconsolidation of object recognition memory[22]. However, the contributions of G protein and β-arrestin biased β-AR signaling in social memory remain largely unknown.

Given the above considerations, the present study focused on the roles of LC-mPFC noradrenergic projections and β-ARs and their downstream signaling pathways in SRM consolidation.

## Results

**LC → mPFC NE projections bidirectionally regulate SRM consolidation.** To determine the effect of LC-mPFC NE projections on social memory storage, we applied a well-validated three-chamber social recognition memory task[23]. We expressed eNpHR3.0-EYFP or ChR2-mCherry in LC NE neurons in TH-Cre mice and detected NE terminals in the mPFC (EYFP+ or mCherry+, Fig. 1a, b, g, h). Optogenetic inhibition of the eNpHR3.0+ NE terminal[24] in the mPFC significantly decreased NE release, while optogenetic activation of the ChrimsonR+ NE terminal[24] in the mPFC significantly increased NE release

(Supplementary Fig. 1a–d). In the habituation session, which include the initial exposure to the three-chamber apparatus, all the mice showed a similar preference for the two outer chambers. During the training phase (sociability test), the mice spent significantly more time interacting with a novel mouse (N) over the empty wire cage (E) (Supplementary Fig. 1e, f), suggesting similar sociability between the two groups. We selectively stimulated LC-mPFC NE projections immediately after the sociability test (Fig. 1a, g). The SRM test was performed 1 hour or 1 day after the mice were exposed to the novel mouse, and it involved comparing the interaction time with the now familiar mouse (F) to the interaction with a second novel mouse (N). Optogenetic inhibition of LC-mPFC NE projections[25] did not change the preference for the novel mouse in SRM test 1 (Fig. 1c, d); however, it decreased the discrimination index in SRM test 2 (Fig. 1e, f), suggesting that LC-mPFC NE projections are required for long-term SRM but not short-term SRM. Furthermore, in the three-chamber SRM task, we found that optogenetic activation of LC-mPFC NE projections after training did not increase exploration for the novel mouse in the SRM test conducted 1 day later (Fig. 1i, j). SRM fades quickly and persists for only days in mice[26]. When the SRM test was performed 4 days after training, the control mice explored the novel and familiar mice almost equally, while the mice with activation of LC-mPFC NE projections showed a considerably greater preference for the novel mouse than that of the control group (Fig. 1k, l), suggesting that the enhancement of NE release in the mPFC prolonged social memory maintenance and promoted SRM consolidation. The activation of β-adrenergic receptors recruits extracellular signal-regulated kinase (ERK), facilitating long-term potentiation maintenance and long-term memory formation[27]. Thus, we examined pERK levels in the mPFC after the sociability test with laser stimulation. We found that exposure to a novel mouse significantly increased pERK levels in the mPFC compared to the mice with exposure to a familiar mouse or the empty wire cage (Supplementary Fig. 2). Moreover, inhibition of LC-mPFC NE projections suppressed ERK activation in the mPFC after the sociability test (Fig. 1m, n). Activation of LC-mPFC NE projections increased ERK activation in the mPFC of naive mice but did not enhance ERK activation further after the sociability test (Fig. 1m, o). These findings indicate that LC-mPFC NE projections might regulate SRM consolidation through adrenergic downstream signaling, such as ERK activation.

**β-arrestin-biased β-AR signaling pathway in the mPFC mediates SRM consolidation.** We selectively knocked out *Adrb1* or *Adrb2* in the mPFC glutamatergic neurons by injecting $AAV_9$-*mCaMKIIα-EGFP-P2A-iCre* into the mPFC of *Adrb1fl/fl* or *Adrb2fl/fl* mice (Fig. 2a–f). In the three-chamber SRM task, mice with half or complete deletion of β1-AR in the mPFC showed a decreased preference for the novel mouse in the SRM test (Fig. 2g–i), while the mice with complete deletion, but not half deletion, of β2-AR in the mPFC showed an impaired preference for the novel mouse in the SRM test (Fig. 2j). All the mice preferred the novel mouse to the empty cage during the sociability test (Supplementary Fig. 3a, b), suggesting that deletion of β1-AR and β2-AR in the mPFC did not impair sociability. Selective knockout of β1-AR in the mPFC did not affect locomotor activity (Supplementary Fig. 3c); however, it did increase anxiety levels, as evidenced by the decreased distance in the central area, and decreased time spent in the central area during the open-field task and the light side during the L/D box task (Supplementary Fig. 3d–h). Selective knockout of β2-AR in the mPFC did not have a significant impact on locomotion or anxiety levels (Supplementary Fig. 3i–n). These results showed that the mice with

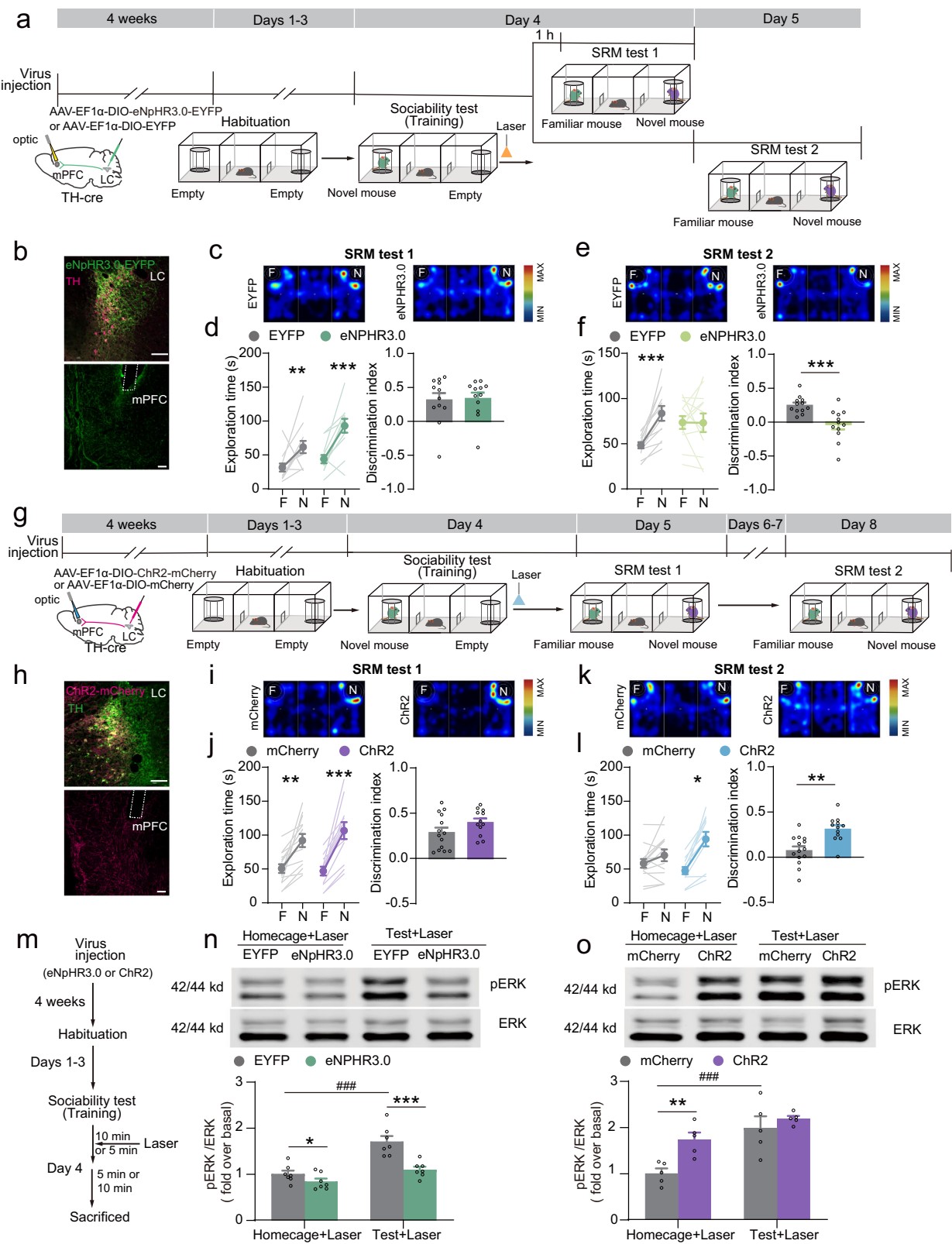

β1-AR or β2-AR knockout in the mPFC were unable to demonstrate social memory, indicating that mPFC β-AR expression is required for SRM consolidation.

According to the above results, we tested the effects of the nonselective β-AR antagonist propranolol and carvedilol on SRM consolidation. When propranolol was bilaterally infused into the mPFC immediately after the sociability test, long-term SRM was

significantly impaired (Fig. 3a–c), while short-term SRM remained intact (Supplementary Fig. 4a–c). In addition, propranolol infusion into the mPFC after training did not impair memory consolidation of fear conditioning (Supplementary Fig. 5). In contrast to the effects of propranolol, carvedilol, a G protein biased β-AR antagonist[28–30], did not affect SRM consolidation (Fig. 3d), suggesting that SRM consolidation might

**Fig. 1 LC-mPFC NE projections regulate SRM consolidation. a, g** Experimental scheme. *AAV₉-EF1α-DIO-eNpHR3.0-EYFP* or *AAV₉-EF1α-DIO-ChR2-mCherry* was injected into the LC of TH-Cre mice and optical fibers were bilaterally implanted over the mPFC. LC-mPFC NE projections were optically stimulated after training and SRM tests were carried out 1 h, 1 day or 4 days after training. Laser (590 nm, 10 mW, constantly for 10 min; 473 nm, 5 mW, twenty 5-ms pulses at 25 Hz, every 5 s for the duration of 5 min) was delivered after sociability test. **b, h** Representative images of eNpHR3.0-EYFP (**b**) or ChR2-mCherry (**h**) expression in the LC with TH immunostaining and optical fiber tip in the mPFC. **c, e, i, k** Heat maps of SRM test. **d, f, j, l** Statistical graph of exploration time for the familiar (F) and novel (N) mice and discrimination scores [EYFP: $n = 12$, eNPHR3.0: $n = 12$. **d** Left: $F_{mouse \times virus}$ $(1, 22) = 1.628$, $p = 0.215$, two-way RM ANOVA; Right: $Z = 70.000$, $p = 0.931$, Mann–Whitney $U$-test. **f** Left: $F_{mouse \times virus}$ $(1, 22) = 9.398$, $p = 0.006$, two-way RM ANOVA; Right: $t (22) = 3.807$, $p < 0.001$, two-tailed Student's $t$ test. mCherry: $n = 14$, ChR2: $n = 12$. **j** Left: $F_{mouse \times virus}$ $(1, 24) = 1.860$, $p = 0.185$, two-way RM ANOVA; Right: $t (24) = -1.664$, $p = 0.113$, two-tailed Student's $t$ test. **l** Left: $F_{mouse \times virus}$ $(1, 24) = 11.904$, $p = 0.002$, two-way RM ANOVA; Right: $t (24) = -3.983$, $p < 0.001$, two-tailed Student's $t$ test]. **m** Experimental scheme. *AAV₉-EF1α-DIO-eNpHR3.0-EYFP* or *AAV₉-EF1α-DIO-ChR2-mCherry* was injected into the LC of TH-Cre mice. LC-mPFC NE projections were optically stimulated after sociability test. 15 min after sociability test, pERK levels in the mPFC were examined. **n** Representative western blots and bar graph for pERK levels in the mPFC with inhibition of LC-mPFC NE projections after sociability test [$n = 7$ for each group. $F_{session \times virus}$ $(1, 24) = 6.863$, $p = 0.015$, two-way ANOVA]. **o** Representative western blots and bar graph for pERK levels in the mPFC with activation of LC-mPFC NE projections after sociability test [$n = 5$ for each group. $F_{session \times virus}$ $(1, 16) = 2.807$, $p = 0.113$, two-way ANOVA]. $*p < 0.05$, $**p < 0.01$, $***p < 0.001$ and $###p < 0.001$ vs indicated group. Scale bar: 100 µm.

not depend on the G protein pathway. In the SRM tasks, all the mice showed a significant preference for the novel mouse over the empty cage during the sociability test (Supplementary Fig. 4b, d). These data suggest that β-AR activation in the mPFC is selectively involved in SRM consolidation, and that non-G protein-dependent signaling may be required for this memory process.

We propose that the β-arrestin signaling pathway in the mPFC might be involved in SRM consolidation. To test this hypothesis, we injected *AAV₉-mCaMKIIα-EGFP-P2A-iCre* into the mPFC of *Arrb2^{fl/fl}* mice and selectively knocked out β-arrestin2 in mPFC excitatory neurons (Fig. 3e, f). β-Arrestin2 selective knockout in the mPFC significantly decreased the preference for the novel mouse over the familiar mouse without influencing sociability, anxiety levels, or locomotor activity (Fig. 3h and Supplementary Fig. 6a–g), suggesting that β-arrestin2 expression in the mPFC is required for SRM consolidation. β-Arrestins, which are downstream of GPCRs, act as signal transducers and mediate the activation of a diverse array of signaling and cellular responses, including ERK activation[30–32]. Therefore, we examined ERK activation in the mPFC of wild type (WT) and β-arrestin2 knockout mice after the sociability test. The results showed that the sociability test significantly increased pERK levels in the mPFC of WT littermates, but not β-arrestin2 mPFC knockout mice (Fig. 3g). These results indicate that β-arrestin-mediated signaling pathway plays a crucial role in SRM consolidation.

**LC-mPFC NE projections regulate SRM consolidation through the β-arrestin-biased β-AR signaling pathway.** To confirm that NEergic consolidation of SRM is mediated by β-ARs and β-arrestin-biased signaling pathway in the mPFC, we performed the SRM task with β1-AR and β-arrestin2 mPFC knockout mice with optically stimulation of LC NE terminals in the mPFC (Fig. 4a, b). β1-AR selective knockout in the mPFC impaired SRM consolidation, which was not rescued by optogenetic activation of LC-mPFC NE projections (Fig. 4c). Moreover, impaired SRM consolidation due to selective β-arrestin2 deletion in the mPFC was not restored by activation of LC-mPFC NE projections (Fig. 4e). In addition, the WT mice with activated LC-mPFC NE projections showed persistently higher preferences for the novel mouse than the mice in the control group, while the mice with deletion of β1-AR or β-arrestin2 in the mPFC showed impaired discrimination of the novel mouse 4 day after training even with activation of LC-mPFC NE projections (Fig. 4d, f).

Studies show that high titers of AAV1 exhibit anterograde transsynaptic spread[33]. Injections of the Cre/FlpO recombinase-expressing AAV1 into the brain area containing presynaptic neurons and simultaneously AAV with a Cre/FlpO-inducible expression cassette into the downstream area, allow the selective labelling of postsynaptic neurons innervated by the presynaptic region[34,35]. Then, we injected anterograde *scAAV₁-hSyn-FlpO* with high titer into the LC and *AAV₉-EF1α-fDIO-Cre-mCherry* into the mPFC of *Arrb2^{fl/fl}* mice, which allowed expression of Cre recombinase and then knockout of β-arrestin2 in mPFC neurons innervated by the LC adrenergic and non-adrenergic neurons (Fig. 4g). Mice with selective β-arrestin2 deletion showed a decreased preference for the novel mouse, suggesting that β-arrestin2 expression in the LC-mPFC circuit is required for SRM consolidation (Fig. 4h). All the mice showed intact sociability (Supplementary Fig. 7). The above results indicate that the regulation of LC-mPFC NE release during SRM consolidation is mediated by β-arrestin-biased β-adrenergic signaling.

We developed an adeno-associated virus vector and delivered the R165L mutant rM3Dq (rM3Darr)[36] to the mPFC to pharmacologically activate β-arrestin-dependent signaling with CNO treatment. We first confirmed the activation of β-arrestin-dependent signaling in N2a cells transfected with rM3Darr. The results showed that CNO (1 µM) significantly increased pERK levels 15–30 min after treatment (Fig. 5a). Furthermore, β-arrestin2 knockdown inhibited ERK activation induced by CNO treatment (1 µM) in N2a cells with rM3Darr expression (Fig. 5b), suggesting that rM3Darr activation induces β-arrestin-dependent ERK activation. Then, we expressed rM3Darr-mCherry in the mPFC and performed a three-chamber SRM experiment (Fig. 5c, d). The mice treated with CNO (1 mg/kg, i.p.) immediately after the sociability test did not show a further increased preference for the novel mouse during the SRM test conducted 1 day later; however, they showed a persistent preference for the novel mouse during the SRM test conducted 4 days later (Fig. 5e, f), suggesting that activation of β-arrestin2 signaling promotes SRM consolidation. Next, we expressed rM3Darr-mCherry in the mPFC and eNpHR3.0-EYFP in LC NE neurons and implanted optical fibers in the mPFC (Fig. 5g). Optogenetic inhibition of LC-mPFC NE projections decreased the preference for the novel mouse, while activation of β-arrestin signaling significantly enhanced the preference for the novel mouse (Fig. 5h), suggesting that activation of β-arrestin signaling pathway restored the impaired social memory caused by inhibiting LC-mPFC NE projections. In addition, we simultaneously expressed rM3Darr-mCherry and deleted β1-AR in the mPFC (Fig. 5i). Although SRM consolidation was impaired by β1-AR deletion in the mPFC, activation of β-arrestin signaling pathway restored memory for the familiar mouse and increased the preference for the novel mouse (Fig. 5j). All the mice preferred the novel mouse to the empty cage during the sociability test (Supplementary Fig. 8). These results suggest that SRM consolidation can be promoted by pharmacological activation of β-arrestin-biased signaling.

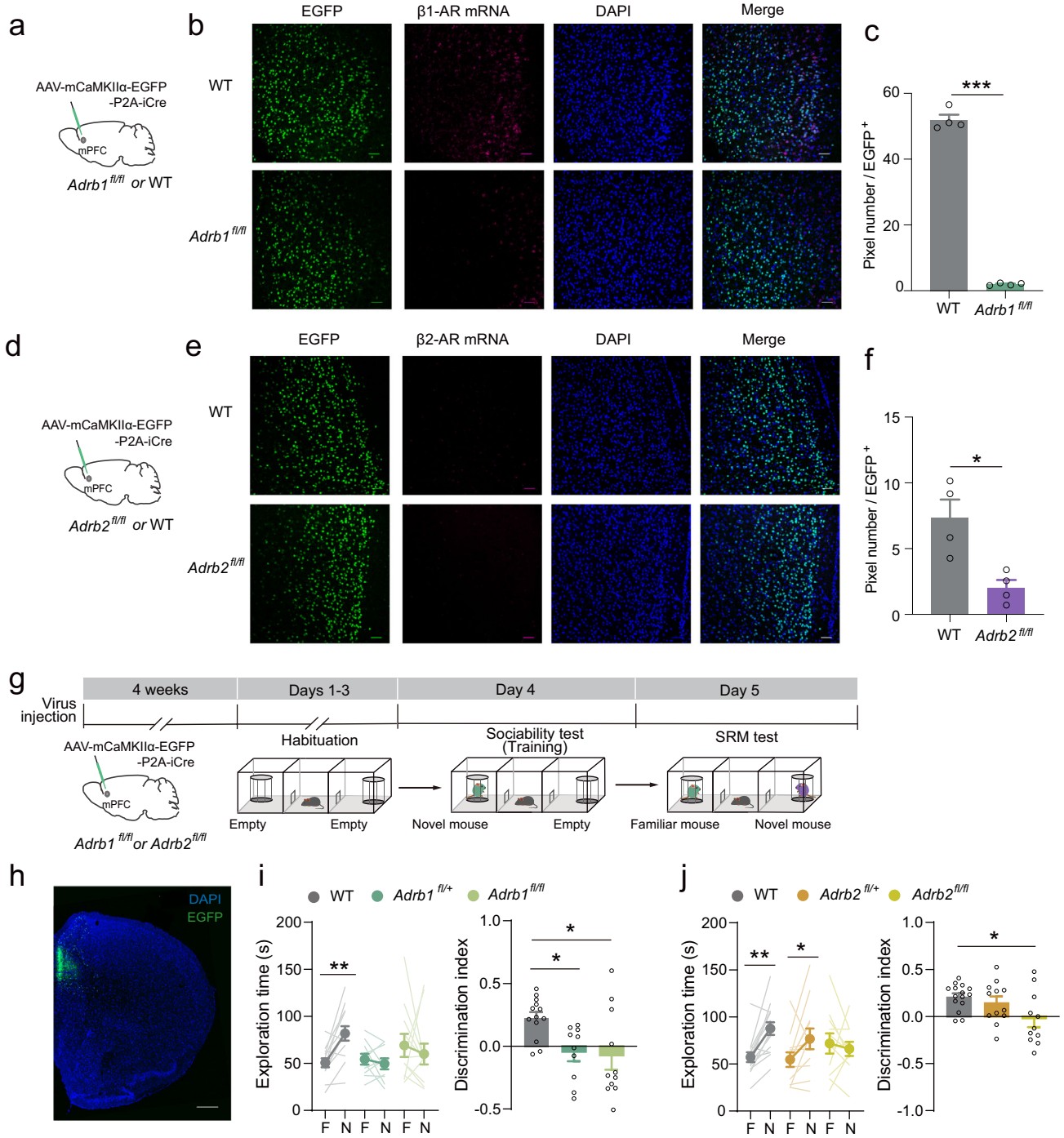

**Fig. 2 β-AR signaling in the mPFC mediates SRM consolidation. a, d** *AAV₉-mCaMKIIα-EGFP-P2A-iCre* was injected in the mPFC of *Adrb1ᶠˡ/ᶠˡ* or *Adrb2ᶠˡ/ᶠˡ* mice and their WT littermates. **b, e, c, f** Representative images and statistical graphs of the *adrb1* and *adrb2* mRNA levels in the mPFC by FISH. [**c** WT: $n = 4$ (3729 cells), *Adrb1ᶠˡ/ᶠˡ*: $n = 4$ (3503 cells), $t(6) = 31.54$, $p < 0.001$, two-tailed Student's $t$ test; **f** WT: $n = 4$ (3192 cells), *Adrb2ᶠˡ/ᶠˡ*: $n = 4$ (3538 cells), $t(6) = 3.543$, $p = 0.012$, two-tailed Student's $t$ test]. Scale bar: 50 μm. **g** Experimental scheme. *AAV₉-mCaMKIIα-EGFP-P2A-iCre* was injected in the mPFC of *Adrb1ᶠˡ/ᶠˡ* or *Adrb2ᶠˡ/ᶠˡ* mice and their WT and heterozygote littermates. Four weeks later, SRM task was performed. **h** Representative image of Cre-EGFP expression in the mPFC. Scale bar: 500 μm. **i, j** Statistical graphs of exploration time for the familiar (F) and novel (N) mice and discrimination scores of the mice with *Adrb1* or *Adrb2* knockout in the mPFC. [**i** WT: $n = 13$, *Adrb1ᶠˡ/⁺*: $n = 10$, *Adrb1ᶠˡ/ᶠˡ*: $n = 11$. $F_{\text{mouse} \times \text{genotype}}(2, 31) = 4.085$, $p = 0.027$, two-way RM ANOVA; Right: $F(2, 31) = 5.364$, $p = 0.01$, one-way ANOVA. **j** WT: $n = 15$, *Adrb2ᶠˡ/⁺*: $n = 12$, *Adrb2ᶠˡ/ᶠˡ*: $n = 11$. Left: $F_{\text{mouse} \times \text{genotype}}(2, 35) = 4.433$, $p = 0.019$, two-way RM ANOVA, Right: $F(2, 35) = 4.139$, $p = 0.024$, one-way ANOVA]. $*p < 0.05$, $**p < 0.01$ and $***p < 0.001$ vs indicated group.

**Pharmacological activation of β-arrestin-biased signaling improves SRM consolidation in aged mice.** A decline in declarative learning and memory performance is a common occurrence associated with aging[37]. Our results show that SRM consolidation is impaired in aged mice (>18 months old, Fig. 6a,

b). Thus, we expressed rM3Darr-mCherry in the mPFC of aged mice (Fig. 6c). The data showed that CNO (1 mg/kg, i.p.) treatment after the sociability test significantly increased the preference for the novel mouse in the SRM test (Fig. 6d). All the mice preferred the novel mouse to the empty cage during the

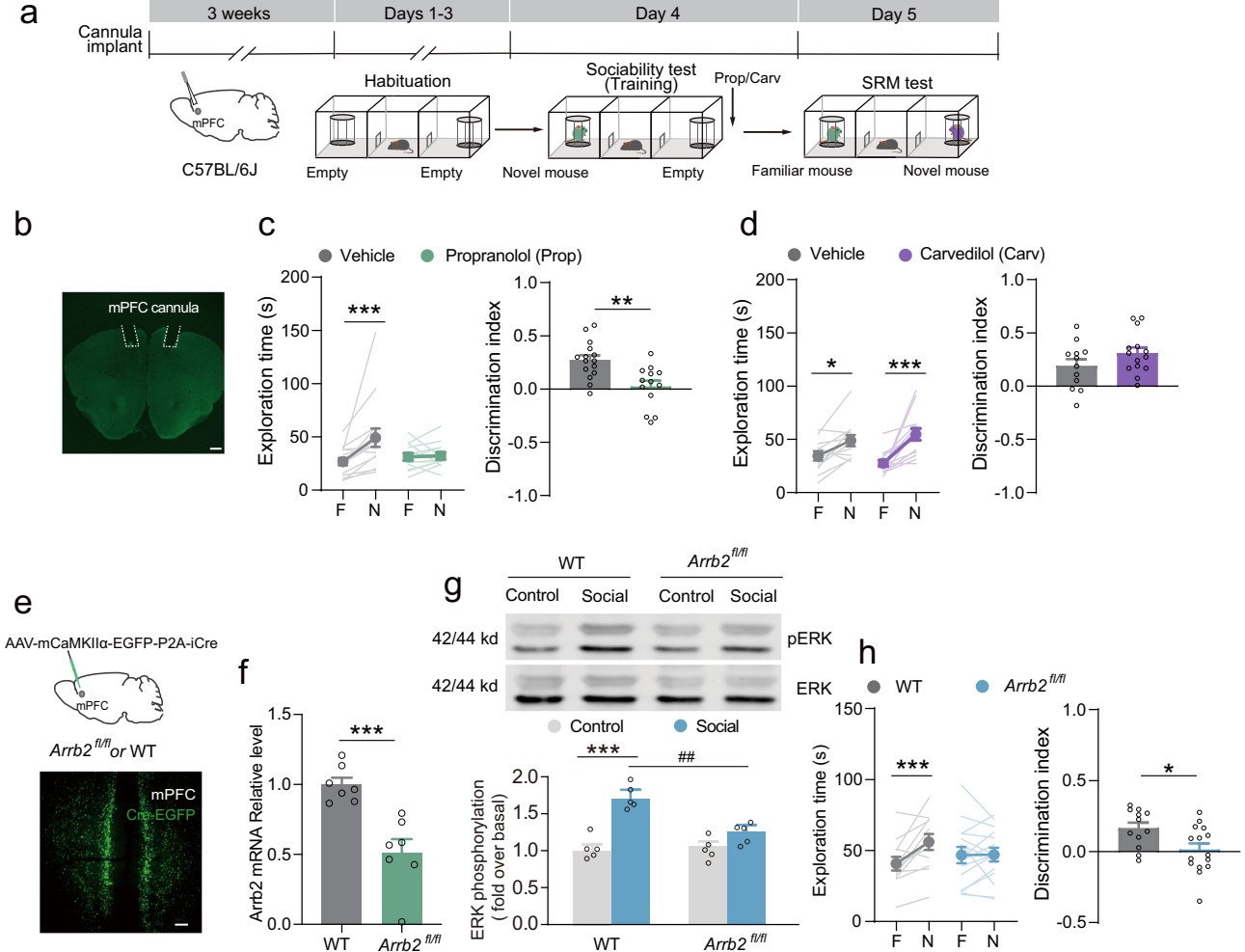

**Fig. 3 β-AR/β-arrestin2 signaling in the mPFC mediate SRM consolidation. a** Experimental scheme. β-AR antagonist, propranolol (10 μg), or carvedilol (5 μg) was infused right after training and SRM tests were carried out 1 day later. **b** Representative image of cannula implantation in the mPFC. Scale bar: 500 μm. **c, d** Statistical graphs of exploration time for the familiar (F) and novel (N) mice and discrimination scores [**c**, Vehicle: $n = 15$, Propranolol: $n = 13$. Left: $F_{\text{mouse} \times \text{treatment}}$ (1, 26) = 6.906, $p = 0.014$, two-way RM ANOVA; Right: $t$ (26) = 3.507, $p = 0.002$, two-tailed Student's $t$ test. **d** Vehicle: $n = 12$, Carvedilol: $n = 14$. Left: $F_{\text{mouse} \times \text{treatment}}$ (1, 24) = 2.707, $p = 0.113$, two-way RM ANOVA; Right: $t$ (24) = −1.477, $p = 0.153$, two-tailed Student's $t$ test]. **e** Representative image of Cre-EGFP expression in the mPFC. $AAV_9$-mCaMKIIα-EGFP-P2A-iCre was injected in the mPFC of $Arrb2^{fl/fl}$ mice and their WT littermates. Scale bar: 100 μm. **f** Bar graph for $Arrb2$ mRNA relative expression level. 4 weeks after injection $AAV_9$-mCaMKIIα-EGFP-P2A-iCre into mPFC of $Arrb2^{fl/fl}$ mice and their WT littermates [$n = 7$ for each group, $t$ (12) = 4.515, $p < 0.001$, two-tailed unpaired $t$ test]. **g** Representative Western blots and bar graph for pERK levels in the mPFC with β-arrestin2 knockout [$n = 5$ for each group. $F_{\text{mouse} \times \text{session}}$ (1, 16) = 11.556, $p = 0.004$, two-way ANOVA]. **h** Statistical graphs of exploration time for the familiar (F) and novel (N) mice and discrimination scores. [WT: $n = 12$, $Arrb2^{fl/fl}$: $n = 15$. Left: $F_{\text{mouse} \times \text{genotype}}$ (1, 25) = 7.094, $p = 0.013$, two-way RM ANOVA; Right: $t$ (25) = 2.557, $p = 0.017$, two-tailed unpaired $t$ test]. *$p < 0.05$, **$p < 0.01$, ***$p < 0.001$ and ##$p < 0.01$ vs indicated group.

sociability test (Supplementary Fig. 9). These data suggest that β-AR/β-arrestin-biased signaling could be a potential drug target for improving social memory in older individuals.

## Discussion
In this study, we found that LC-mPFC NE projections controlled SRM consolidation through β-AR/β-arrestin-biased signaling in the mPFC (Fig. 6e). The activation of LC-mPFC NE projections or β-arrestin signaling promoted social memory maintenance. In addition, the impairment of SRM consolidation caused by NE projection inhibition or β1-AR deletion in the mPFC was restored by activating β-arrestin-biased signaling.

The three-chamber task is a widely used test paradigm for quantitatively measuring sociability and social memory[38,39]. The mouse chooses to enter one of the three chambers; thus, sociability may be measured by directly comparing the preference for a

novel mouse to the preference for an empty wired cage. With this task, social recognition memory can also be evaluated by comparing interactions with a novel mouse to interactions with familiar mice in each chamber. However, it remains unclear whether mice prefer social novelty or just novelty in the wired cage. It is possible that the mice might only care that the previously empty cage now contains something or a novel mouse.

Our data showed that injecting propranolol, a β-adrenergic antagonist, or knocking out β-AR in the mPFC significantly impaired SRM consolidation, while injecting carvedilol, a nonselective β-AR G protein-biased antagonist and α1-AR antagonist, induced no such impairment. The distinct effects of propranolol and carvedilol suggest that β-AR, but not α1-AR, in the mPFC participates in the process of SRM consolidation. Many studies have shown that α2-AR in the mPFC is involved with the working memory, reaching different conclusions. Some studies

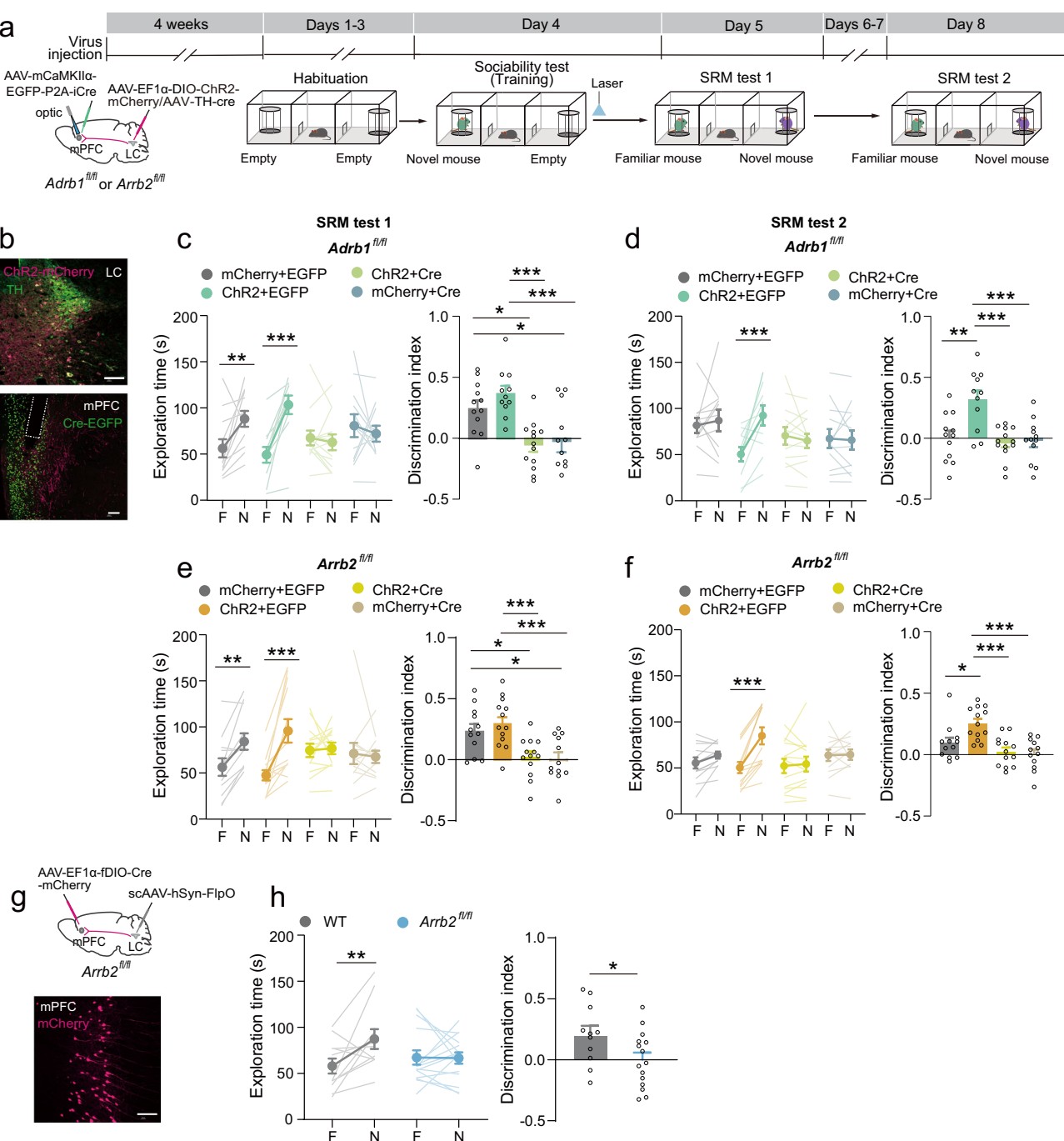

**Fig. 4 Activation of LC-mPFC NE projections did not rescue the impairment of SRM consolidation by β1-AR or β-arrestin2 knockout. a** Experimental scheme. *AAV₉-EF1α-DIO-ChR2-mCherry* and *AAV₉-THP-iCre* were injected in the LC, *AAV₉-mCaMKIIα-EGFP-P2A-iCre* was injected in the mPFC, and optical fibers were implanted above the mPFC of *Adrb1ᶠˡ/ᶠˡ* or *Arrb2ᶠˡ/ᶠˡ* mice. Four weeks later, SRM task was performed. For stimulation of NE terminals in the mPFC, blue light was delivered (473 nm, twenty 5-ms pulses at 25 Hz, every 5-s, 5-min duration) after sociability test and SRM tests were performed 1 day later. **b** Representative images of ChR2-mCherry expression in the LC, Cre-EGFP expression, ChR2-mCherry⁺ terminals, and optical fiber tip in the mPFC. **c–f** Statistical graphs of exploration time for the familiar (F) and novel (N) mice and discrimination scores [**c**, **d** mCherry/EGFP, $n = 12$; ChR2/EGFP, $n = 11$; ChR2/Cre, $n = 13$; mCherry/Cre, $n = 12$. **c** Left: $F_{mouse \times virus}$ (3, 44) = 7.961, $p < 0.001$, two-way RM ANOVA; Right: $F$ (3, 44) = 4.139, $p < 0.001$, one-way ANOVA; **d** Left: $F_{mouse \times virus}$ (3, 44) = 5.895, $p = 0.004$, two-way RM ANOVA; Right: $F$ (3, 44) = 8.928, $p < 0.001$, one-way ANOVA. **e**, **f** mCherry/EGFP, $n = 12$; ChR2/EGFP, $n = 13$; ChR2/Cre, $n = 13$; mCherry/Cre, $n = 12$. **e** Left: $F_{mouse \times virus}$ (3, 46) = 7.304, $p < 0.001$, two-way RM ANOVA; Right: $F$ (3, 46) = 7.469, $p < 0.001$, one-way ANOVA; **f** Left: $F$ mouse × virus (3, 46) = 10.826, $p < 0.001$, two-way RM ANOVA; Right: $F$ (3, 46) = 9.586, $p < 0.001$, one-way ANOVA]. **g** sc*AAV₁-hSyn-FlpO* was injected in the LC and *AAV₉-EF1α-fDIO-Cre-mCherry* was injected in the mPFC of *Arrb2ᶠˡ/ᶠˡ* mice. Representative images of Cre-mCherry expression in the mPFC. **h** Statistical graphs of exploration time for the familiar (F) and novel (N) mice and discrimination scores [WT: $n = 11$; *Arrb2ᶠˡ/ᶠˡ*: $n = 15$. Left: $F_{mouse \times virus}$ (1, 24) = 4.7, $p = 0.04$, two-way RM ANOVA, Right: $t$ (24) = 2.196, $p = 0.038$, two-tailed Student's $t$ test]. *$p < 0.05$, **$p < 0.01$ and ***$p < 0.001$ vs indicated group. Scale bar: 100 μm.

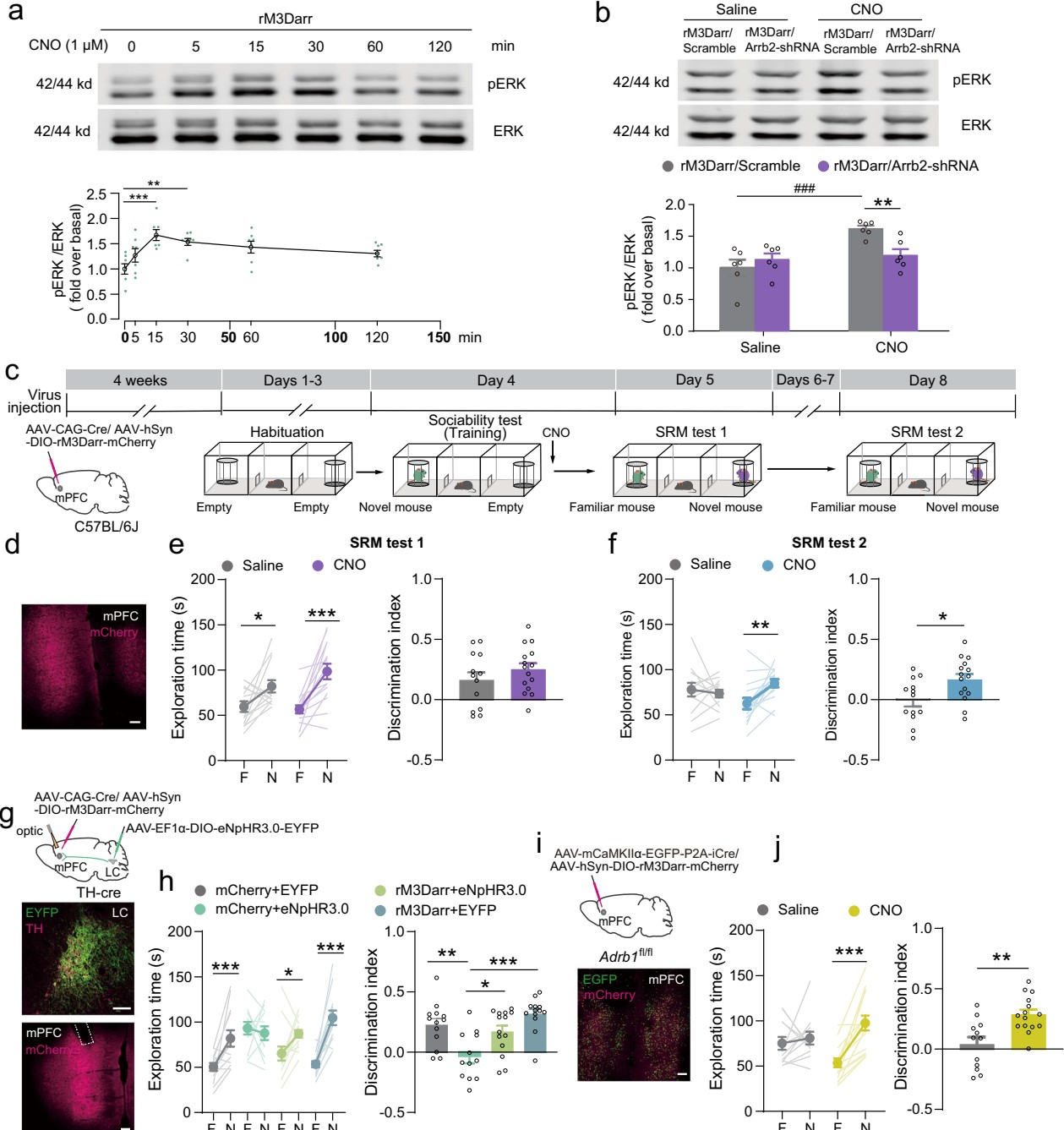

**Fig. 5 Activation of β-arrestin biased signaling in the mPFC rescued impairment of SRM consolidation by LC-mPFC NE projection inhibition or β-AR knockout. a** CNO (1 μM) significantly increased the pERK levels 15–30 min after treatment in N2a cells transfected with rM3Darr [$n = 7$ for each group, 15 min: $F (5, 36) = 5.261$, $p < 0.001$, one-way ANOVA]. **b** *Arrb2* knockdown inhibited ERK activation by CNO treatment (1 μM) in N2a cells transfected with rM3Darr at 15 min [$n = 6$ for each group, $F_{shRNA \times treatment} (1, 20) = 8.207$, $p = 0.010$, two-way ANOVA]. **c** Experimental scheme. *AAV$_9$-hSyn-DIO-rM3Darr-mCherry* and *AAV$_9$-CAG-Cre* were injected in the mPFC of C57BL/6 J mice. Four weeks later, SRM task was performed. The mice were injected with CNO (1 mg/kg, i.p.) after training and SRM tests were performed 1 day and 4 days later. **d** Representative image of rM3Darr-mCherry expression in the mPFC. **e, f** Statistical graphs of exploration time for the familiar (F) and novel (N) mice and discrimination scores [Saline, $n = 13$; CNO, $n = 15$. **e** Left: $F_{mouse \times treatment} (1, 26) = 2.29$, $p = 0.142$, two-way RM ANOVA; Right: $t (26) = -1.121$, $p = 0.273$, two-tailed Student's $t$ test. **f** Left: $F_{mouse \times treatment} (1, 26) = 7.024$, $p = 0.014$, two-way RM ANOVA; Right: $t (26) = -2.579$, $p = 0.016$, two-tailed Student's $t$ test]. **g** Virus injection and representative images of eNpHR3.0-EYFP expression in the LC, rM3Darr-mCherry expression and optical fiber tip in the mPFC. *AAV$_9$-EF1α-DIO-eNpHR3.0-EYFP* was injected in the LC, *AAV$_9$-hSyn-DIO-rM3Darr-mCherry* and *AAV$_9$-CAG-Cre* were injected into the mPFC of TH-Cre mice. Optical fibers were implanted above the mPFC. **h** Statistical graphs of exploration time for the familiar (F) and novel (N) mice and discrimination scores [EYFP/mCherry, $n = 13$; EYFP/rM3Darr, $n = 12$; eNpHR3.0/rM3Darr, $n = 14$; eNpHR3.0/mCherry, $n = 13$. Left: $F_{mouse \times virus} (3, 48) = 7.135$, $p < 0.001$, two-way RM ANOVA, Right: $F (3, 48) = 8.697$, $p < 0.001$, one-way ANOVA]. **i** Virus injection and representative image of rM3Darr-mCherry and Cre-EGFP expression in the mPFC. **j** Statistical graphs of exploration time and discrimination scores [Saline, $n = 12$; CNO, $n = 15$. Left: $F_{mous \times treatment} (1, 25) = 9.238$, $p = 0.005$, two-way RM ANOVA; Right: $t (25) = -3.638$, $p = 0.001$, two-tailed Student's $t$ test]. *$p < 0.05$, **$p < 0.01$, ***$p < 0.001$ and $^{###}p < 0.001$ vs indicated group. Scale bar: 100 μm.

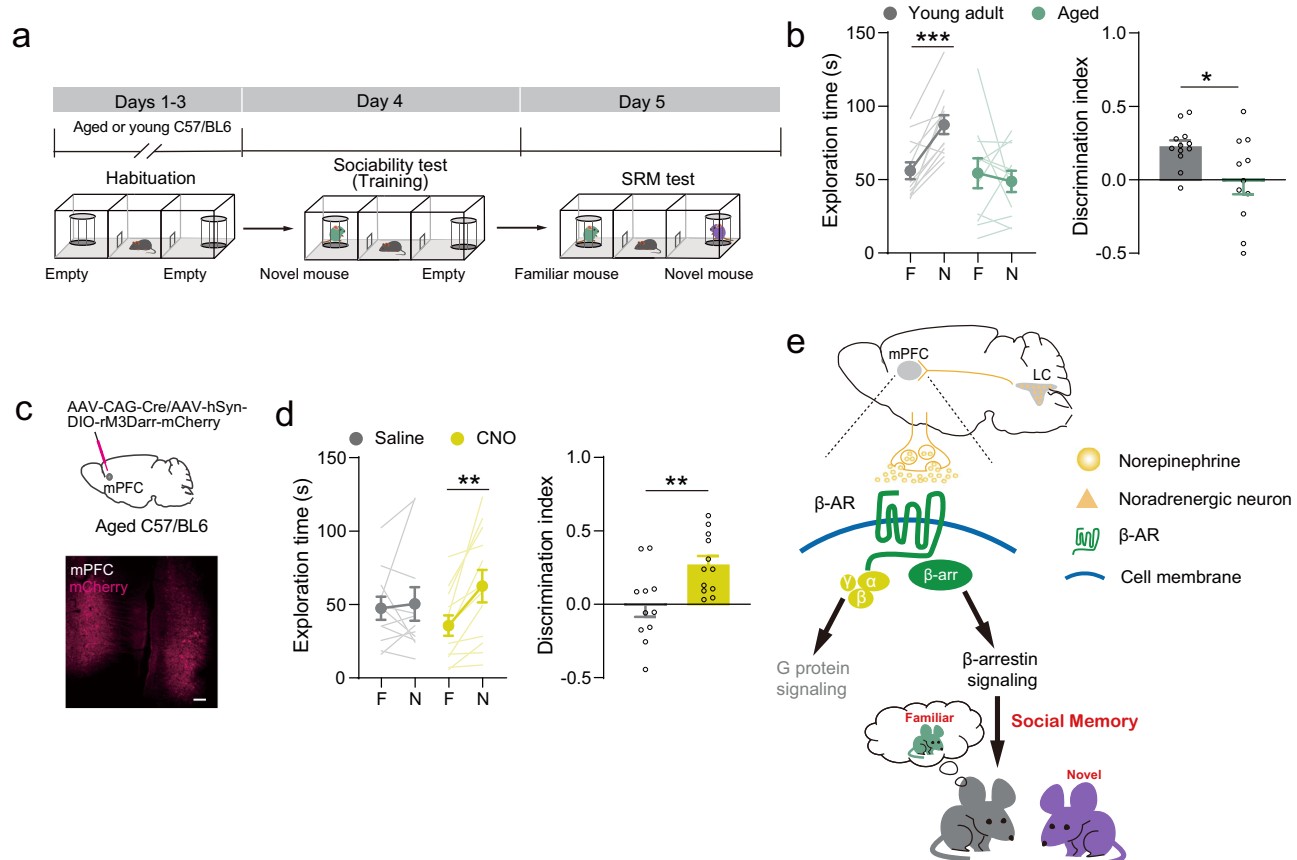

**Fig. 6 Pharmacological activation of β-arrestin biased signaling improved SRM consolidation in aged mice. a** Experimental scheme. SRM task was carried out in aged mice (>18 months old). **b** Statistical graphs of exploration time and discrimination scores [Young adult, $n = 12$; Aged, $n = 11$. Left: $F_{mouse \times age}$ (1, 21) = 9.816, $p = 0.005$, two-way RM ANOVA; Right: $Z = 33.000$, $p = 0.045$, Mann–Whitney U-test]. **c** Representative image of rM3Darr-mCherry expression in the mPFC. *AAV₉-hSyn-DIO-rM3Darr-mCherry* and *AAV₉-CAG-Cre* were injected into the mPFC of aged mice (>18 months old). **d** Statistical graphs of exploration time for the familiar (F) and novel (N) mice and discrimination scores [Saline, $n = 11$; CNO, $n = 12$. Left: $F_{mouse \times treatment}$ (1, 21) = 4.979, $p = 0.037$, two-way RM ANOVA, Right: $t$ (21) = −2.95, $p = 0.008$, two-tailed Student's $t$ test]. **e** Working model illustrating that LC-mPFC NE projections regulate memory consolidation of social recognition through β-arrestin2-biased signaling. *$p < 0.05$, **$p < 0.01$ and ***$p < 0.001$ vs indicated group. Scale bar: 100 μm.

have reported that α2-AR activation impairs mPFC functions, such as working memory performance[37,40]. However, other studies have reported that α2 adrenergic agonists improve the functions of the prefrontal cortex, including working memory[41,42]. In addition, the administration of an α2-adrenergic antagonist (which increases NE concentrations) has been shown to improve social recognition[43]. Due to the differential expressions of α and β adrenergic receptors in pyramidal neurons and interneurons, the different roles of α-AR and β-AR in the mPFC in social recognition memory consolidation needs to be studied further.

Upon ligand binding, G protein–coupled receptors (GPCRs), including β-ARs, undergo conformational changes that promote their binding to heterotrimeric G proteins and β-arrestins. Recent studies have indicated that heterotrimeric G proteins and β-arrestins are both capable of independently interacting with and recruiting intracellular signaling molecules[44,45]. Most ligands that bind to GPCRs have balanced or unbiased signaling activity through G proteins and β-arrestins. Some ligands with a preference for one pathway over the other pathways have also been discovered. Carvedilol is a G protein-biased β-AR antagonist that mildly activates β-arrestin signaling. Propranolol is a full antagonist that blocks both the G protein and β-arrestin pathways. Previous studies have shown that carvedilol selectively promotes the recruitment of Gαi to β1-AR, activating the β-

arrestin2 biased pathway[28,46], and inducing β1-AR-mediated transactivation of the EGFR and β-arrestin2-dependent ERK activation[30], suggesting that β-arrestin2 can mediate β1-AR signaling independently. It has long been posited that the conventional G protein/cAMP/PKA signaling pathway mediates the role of β-ARs in memory. Prior studies have shown that blocking β-AR transmission with propranolol impairs memory consolidation or reconsolidation during fear conditioning, object recognition, and cocaine-induced CPP[42,47,48]. The inhibition of the activities of certain components of G protein-coupled pathways, such as PKA and CREB, disrupts memory consolidation and reconsolidation[49,50]. However, memory reconsolidation requires protein synthesis but not PKA activation[51]. Our previous studies have suggested that β-arrestin biased signaling in the entorhinal cortex mediates the reconsolidation of object recognition memory[22], suggesting that the β-arrestin dependent signaling pathway should play a role in memory storage. In this study, carvedilol did not suppress SRM in the memory retention test 1 while propranolol did, suggesting that SRM consolidation after training might not depend on a G protein-biased pathway. Furthermore, mice with β-arrestin 2 knockout in the mPFC showed impaired SRM consolidation. When β-arrestin signaling pathway was activated after training, SRM maintenance was promoted, and the impaired SRM consolidation caused by the inhibition of LC-mPFC projections or β1-AR deletion was rescued. Thus, we

propose that the regulation of LC-mPFC NE projections might regulate SRM consolidation through β-arrestin-biased signaling pathways in the mPFC.

Previous studies have indicated that aged animals show significantly reduced interaction levels with new juvenile mice during social tasks, even with the absence of overt cognitive decline[52,53], suggesting that social circuits are particularly vulnerable during aging. It has long been reported that LC neurons are significantly reduced in older people[54] and animals than in young individuals[55], which might contribute to social memory issues in aged mice. In this study, we found that the direct activation of β-arrestin signaling by stimulating rM3Darr-expressing neurons in the mPFC significantly increased the preferential exploration in aged mice. Thus, our study demonstrates that β-arrestin-biased signaling is critical for SRM storage.

The concept of ligand bias in GPCR downstream signaling has recently gained increasing attention. Several biased ligands have clinical potential, including β-arrestin-biased ligands of the angiotensin II type 1 receptor (AT1R)[56], G-protein-biased ligands of the μ-opioid receptor (MOR)[57], and β-arrestin-biased ligands of the dopamine D2 receptor[58]. However, the number of biased ligands that have been reported in the literature is still limited. Our results suggest that β-AR/β-arrestin-biased signaling is critical for SRM storage, demonstrating the potential for developing novel β-AR agonists with specific β-arrestin-biased activation for memory enhancement.

Our study suggests that LC-mPFC NE projections regulate social recognition memory consolidation through the β-AR/β-arrestin-biased signaling pathways, providing evidence for the neurobiological functions of β-arrestin-biased pathway in memory storage. These data indicate that β-arrestin-biased β-adrenergic ligands may be a potential drug target for improving memory storage and treating psychological diseases.

## Methods

**Animals.** $Adrb1^{fl/fl}$ and $Adrb2^{fl/fl}$ transgenic male mice were developed by our lab and backcrossed more than 10 times to the C57BL/6J strain ($Adrb1$: ENSMUSE000000000024435; $Adrb2$: ENSMUSE00000399288). $Arrb2^{fl/fl}$ were kindly provided by Professor Pei Gang (Shanghai Institutes for Biological Sciences, Chinese Academy of Sciences). TH-Cre mice were purchased from Jackson Laboratory (stock number: 008601). Five-week-old male C57BL/6J mice were purchased from the Shanghai Laboratory Animal Center, CAS. All the mice were group housed with 3–4 per cage under a 12 h light–dark cycle (20:00–8:00 light on) with access to water and food ad libitum. Adult mice (8–10 weeks old) and aged mice (>18 months) were used. All behavioral tests were conducted during their light off period. All animal treatments were strictly in accordance with the National Institutes of Health Guide for the Care and Use of Laboratory Animals and were approved by the Animal Care and Use Committee of Shanghai Medical College of Fudan University. The offsprings were genotyped using the following primer sets: 5'-CTGTTCGGCATCGGAATGAAGC-3'; 5'-TGACGTCATGAACTGGGATTTC AG-3' ($Adrb1^{fl/fl}$ mice); 5'-TTGCCGGCAGTCTGAAGAAGC-3'; 5'-AGGAAGG ATTGTCTCCCAGTATGAC-3' ($Arrb2^{fl/fl}$ mice); 5'-GGTTGCACAGCAGC CCTAGAT-3'; 5'-CCGTTATGTGCACCAGACTTTAGG-3' ($Adrb2^{fl/fl}$ mice); 5'-G AGACAGAACTCGGGACCAC-3'; 5'-AGGCAAATTTTGGTGTACGG-3' (TH-Cre mice). All behavioral subjects were individually habituated to the experimenter at least for 3 days.

**Reagents.** Propranolol [(+/−)-propranolol HCl] (Sigma-Aldrich, #P0884-1G) was dissolved in saline. Carvedilol (Tocris Bioscience, #2685) was dissolved in saline containing 1% dimethylsulfoxide. Propranolol (10 μg) or carvedilol (5 μg) was bilaterally infused in the mPFC. Clozapine N-oxide [CNO] (Sigma-Aldrich, #C0832-5MG) was dissolved in saline. The mice were injected intraperitoneally with CNO (1 mg/kg, i.p.).

**Vector construction and viral vectors.** The pcDNA3.1 vector carrying rM3Darr was provided by Professor Ken-ichiro Nakajima who generated this construct containing M3 with the R165L point mutation. AAV9-hSyn-DIO-rM3Darr-mCherry was developed to selectively activate β-arrestin-biased signaling without perturbing G protein mediated pathways in response to CNO treatment as described previously[36]. The Cre sequence from pCAG-Cre-GFP (Addgene Plasmid # 13776) was cloned into pAAV2-THP-EGFP (Addgene Plasmid # 80336) through ECOR I and Sal I restriction enzyme sites. $AAV_9$-$mCaMKII\alpha$-EGFP-P2A-iCre,

$AAV_9$-$mCaMKII\alpha$-EGFP, $AAV_9$-$EF1\alpha$-Flex-ChrimsonR-tdTomato, $AAV_9$-THP-Cre, scAAV$_1$-hSyn-FlpO, and $AAV_9$-$EF1\alpha$-fDIO-Cre-mCherry were packaged by Taitool Biological Co., Ltd. $AAV_9$-$EF1\alpha$-DIO-eNpHR3.0-EYFP, $AAV_9$-$EF1\alpha$-DIO-EYFP, $AAV_9$-hSyn-DIO-rM3Darr-mCherry, $AAV_9$-hSyn-DIO-mCherry, $AAV_9$-hSyn-NE2h-EGFP, and $AAV_9$-CAG-Cre were packaged by Neuron Biotech Co., Ltd. $AAV_9$-$EF1\alpha$-DIO-hChR2(H134R)-mCherry and $AAV_9$-$EF1\alpha$-DIO-mCherry were packaged from the BrainVTA Co., Ltd. AAV titers ranged from $2.0 \times 10^{12}$ to $2.5 \times 10^{12}$ vector genome (vg) ml$^{-1}$ were used in all experiments. High titer of scAAV$_1$-hSyn-FlpO ($1 \times 10^{13}$ vector genome (vg) ml$^{-1}$) was applied for anterograde infection.

**Stereotaxic surgery.** The mice were anesthetized with isoflurane (3.5% induction, 1.5–2% maintenance), and placed in a stereotaxic apparatus (Stoelting Instruments, USA). The virus was infused in the mPFC (from bregma: anterior-posterior (AP), + 1.8 mm; mediolateral (ML), ± 0.30 mm; and dorsal-ventral (DV), −2.5 mm), or the LC (AP, −5.4 mm; ML, ± 0.85 mm; and DV, −4.0 mm). The virus was delivered using a 10 μl syringe and a 36-gauge blunt needle under the control of a UMP3 ultra micro pump (World Precision Instruments, USA) with a controlled volume and flow rate (150 nl at 50 nl/min). After injection, the needle was left for an additional 10 min and was then slowly removed. The optical fibers (0.37 NA, 200 μm core diameter; Anilab) were implanted above the mPFC (AP, +1.9 mm; ML, ±1.1 mm; and DV, −2.4 mm, 20° angle). Following surgery, the mice were allowed to recover for at least three weeks before behavioral experiments.

**Cannula implantation and drug delivery.** Mice were anesthetized with isoflurane (3.5% induction, 1.5-2% maintenance) and placed in a stereotaxic apparatus. The drug pedestal guide cannulas (27 gauge, RWD Life Science Co. Ltd.) were implanted bilaterally 1 mm above the mPFC (AP, +1.9 mm; ML, ±1.1 mm; and DV, −1.4 mm, 20° angle). Animals were allowed to recover from the surgery for at least 2 weeks before the behavioral tests. A 34-gauge steel needle with a 1.0-mm projection was connected to an infusion pump (BAS Bioanalytical Systems Inc.). Propranolol (10 μg), carvedilol (5 μg), or saline was infused via the guide cannula bilaterally after 3-chamber sociability test or fear conditioning.

**Immunohistochemistry.** Animals were transcardially perfused with 4% paraformaldehyde (PFA) in 0.1 M sodium phosphate buffer (PB) and then further post-fixed for 4 h. Brains were cryoprotected with 30% (wt/vol) sucrose/phosphate-buffered saline (PBS) for at least 48 h and slices were sectioned at 30 μm by a vibratome (CM3050S, Leica). Slices were blocked with 10% normal goat serum in PBS with 0.3% Triton X-100 (PBST), then incubated with primary antibodies against TH[59] (mouse; 1:1000; Millipore AB152), RFP[60] (rabbit, 1:500; Rockland, 600-401-379) or GFP[61] (chicken, 1:500; Thermo Fisher Scientific, A-10262), in PBST at 4 °C overnight. Sections were rinsed three times with PBS for 15 minutes each at room temperature (RT), followed by incubation of secondary antibody [Alexa-488 or Cy3 IgG (mouse, rabbit, or chicken, 1:50,000, Jackson ImmunoResearch)] in PBST at RT for 2 h. Sections were then rinsed three times with PBS for 15 minutes each and then mounted on slides. Images were viewed and photographed with Nikon A1 confocal using ×10 or ×20 objectives.

**Three-chamber SRM task.** The social behavior task was performed in a three-chamber apparatus, a rectangular, non-transparent Plexiglas box (40 cm length × 63 cm width × 23 cm height). The box was divided by two Plexiglas walls with a small circular door (7 cm in diameter). Two outer chambers (40 cm length × 21 cm width × 23 cm height) were connected by a center chamber (40 cm length × 21 cm width × 23 cm height). The SRM test was performed after the Habituation session and Sociability test[10].

*Habituation.* The subject mouse was released in the middle compartment and permitted to explore the three-chambered box with an empty wire container in each outer chamber for 10 min for 3 consecutive days.

*Sociability test (Training).* On the fourth day, a juvenile male mouse (novel mouse, 3–5 weeks old), which had no prior contact with the subject mouse, was introduced into a wired cage in one chamber designated as the "social" compartment. Meanwhile, an identical wire cage that remained empty was placed in the other chamber. The subject mouse was released in the middle chamber and allowed to explore the three chambers for two 10-min trials, with a 15-min interval between trials. The amount of time mice explored within a 2 cm radius proximal to each wire cage (approach, stand, sniffing mounting, and nose-to-nose contact time) was counted.

*Social recognition memory test (SRM test).* SRM tests were carried out 1 h, 1 day, or 4 days after Sociability tests, and the subject mouse was released in the middle chamber for 10 min. During memory retention test, one chamber contained a wire cage with a juvenile stranger male mouse (novel mouse, 3–5 weeks old) and the other chamber contained a wire cage with the familiar mouse previously exposed in sociability test (familiar mouse). Thus, the familiar mouse is the same used in

sociability test and novel mouse is different in each SRM test. Again, the location of novel mouse and familiar mouse was counterbalanced between sessions. The wire cage was cleaned thoroughly before and between trials. All the sessions were taped with a digital video camera. The amount of time that the mice explored the familiar and novel mice during each session was analyzed with Ethovision XT software (Noldus, Wageningen, Netherlands).

The sociability discrimination score (Eq. 1) was calculated as:

$$\left[(\text{time exploring novel mouse} - \text{time exploring empty cage})\right] / \left[(\text{time exploring novel mouse} + \text{time exploring empty cage})\right] \quad (1)$$

The social memory discrimination score (Eq. 2) was calculated as:

$$\left[(\text{time exploring novel mouse} - \text{time exploring familiar mouse})\right] / \left[(\text{time exploring novel mouse} + \text{time exploring familiar mouse})\right] \quad (2)$$

**Optogenetic stimulation for social behavior tests**. Optogenetic stimulation was delivered immediately after the sociability test (3–4 weeks after virus infusion and optic fiber implantation). The optic fibers were connected to a 473 nm (blue light) or 590 nm laser (yellow light) (Shanghai Dream Lasers Technology Co. Ltd.) through a patch cord with a pair of FC/PC connectors and a fiber optic 1 × 2 rotary joint. For stimulation of ChR2, blue light was delivered at 5 mW with twenty 5-ms pulses at 25 Hz, every 5 s for the duration of 5 min. For stimulation of eNpHR3.0, yellow light was delivered constantly at 10 mW for 10 min. Laser light output through the optical fibers was adjusted by a digital power meter console and modulated with a Master 8 pulse stimulator (A.M.P.I.).

**Fiber photometry recording for NE release**. $AAV_9$-$EF1\alpha$-$Flex$-$ChrimsonR$-$tdTo$-$mato$ or $AAV_9$-$EF1\alpha$-$DIO$-$eNpHR3.0$-$EYFP$ and $AAV_9$-$hSyn$-$NE2h$-$EGFP$ virus were individually injected into the LC and the mPFC. An optical fiber (200 μm diameter, 0.48 numerical aperture (NA), Hangzhou Newdoon Technology) was implanted unilaterally into the mPFC. Photometry recordings were conducted after virus infusion and optic fiber implantation 3–4 weeks. For optogenetic photostimulation, the optic fibers were connected 590 nm laser (yellow light) (Thinker Tech Nanjing Biotech) through a patch cord. For stimulation of ChrimsonR+ NE terminals in the mPFC, optical stimulation (590 nm, 5 mW, 5-ms pulses at 5, 25, and 50 Hz, 1 s duration) was delivered every 15 s for 15–20 trials through one optic fiber implanted in the mPFC. For stimulation of eNpHR3.0 terminals in the mPFC, optical stimulation (590 nm, 10 mW, 10 s duration) was given every 15 s for 15–20 trials through one optic fiber implanted in the mPFC. At the same time, fluorescence dynamics of NE sensor, NE2h[62], was recorded through the same optic fiber implanted in the mPFC. Fiber photometry was performed similar to before. Briefly, NE sensor was excited using two excitation sources corresponding to 470 nm wavelength and 405 nm wavelength LED light. The light passed through excitation filters onto an optic fiber patch cable that was connected to the chronically implanted fiber. Light intensity at the tip of the patch cable was around 0.25 mW. NE sensor emission light travelled back through the same fibers onto a photoreceiver. The analog voltage signals were digitalized at 100 Hz and recorded using the software Fiber photometry (Thinker Tech Nanjing Biotech). The data were segmented based on optical stimulation events within individual trials. Z-scores of the 2-s before stimulation were taken as the baseline. Photometry data were analyzed with custom-written MATLAB codes (MATLAB R2019a, MathWorks).

**Fear conditioning**. For fear conditioning task, the mice were introduced into the conditioning chamber (Med Associates) for 3 min. One day later, mice were put back in the chamber and received three pairs of tone-footshock (Tone: 2800 Hz, 85 dB, 30 s; FS: 0.5 mA, 1 s) with a 2-min intertrial interval. Twenty-four hours later, a contextual fear memory test was performed, and the mice were exposed to the conditioning chamber for 3 min without tone. Another cohort of mice was conducted cued fear memory test with 3 trials of tone (2800 Hz, 85 dB, 30 s) with 30 sec intervals in a novel context. The percentage of time spent freezing was automatically analyzed by a computer program (Med Associates).

**Open field (OF) test**. OF test is one of the most frequently used methods to evaluate locomotor activity and innate anxiety levels of rodents. Two days before the tests, mice were allowed to habituate to the environment where the OF test was performed. Locomotor activity was measured in an open field arena (40 × 40 cm) for 30 min under 25 Lux luminance. The apparatus was cleaned before and between trials. Total distance, distance, and duration in the center area were analyzed using an automated detection system (TopScan, CleverSys. lnc.).

**Light/dark box (L/D) test**. The L/D box (46 × 27 × 30 cm) was made of Plexiglas and consisted of two compartments. The larger section was the bright compartment (two-thirds of the box) and the smaller section was the dark compartment (one-third of the box). The test was conducted with 25 Lux luminance in the light box. Mice were released from the center of the lightbox with their back to the entrance and allowed to explore the apparatus for 6 min. The L/D box was cleaned before and between trials. The time spent in the dark compartment was analyzed using an automated detection system (TopScan, CleverSys. lnc.).

**Elevated O maze (EOM) test**. The elevated O maze consisted of two open arms and two closed arms with walls on the side. The height of the O maze was 100 cm from the floor. The mice were placed in the center of the open arm and were allowed to explore for 6 min. The O maze was cleaned before and between each trail. The time spent in each arm was analyzed with Clever System software (TopScan, CleverSys. lnc.).

**Fluorescence in situ hybridization (FISH)**. The mice were perfused intracardially with saline first, then with 4% paraformaldehyde in 0.1 M $Na_2HPO_4$/$NaH_2PO_4$ buffer (pH = 7.5), and the brains were removed. After post-fixation in 4% paraformaldehyde for 4 h, the samples were stored in 30% sucrose/PBS for 3 days. FISH was performed on the fixed frozen brain slices 10 μm thick, following the RNAscope procedures (Advanced Cell Diagnostics, Inc., Newark, CA, USA). In brief, frozen sections (10 μm thick) were cut coronally through the mPFC formation. Sections were thaw-mounted onto Superfrost Plus Microscope Slides (Fisher Scientific, Waltham, USA) and pretreated for protease digestion for 10 min at room temperature. Sections were then incubated with probes of mouse Adrb1 and Adrb2 (Adrb1, accession No: NM_007419.2, target region 158–1830; Adrb2, accession No: NM_007420.3, target region 55–962) for 2 h at 40 °C with labeled probe mixture per slide. The nonspecifically hybridized probe was removed by washing the sections in 1× washing buffer at room temperature, followed by Amplifier 1-FL for 30 min, Amplifier 2-FL for 30 min, and Amplifier 3-FL for 15 min at 40 °C. Each amplifier was removed by washing with 1× washing buffer for 2 min at room temperature. The slides were viewed, analyzed, and photographed with an LSM 510 microscope (Zeiss).

**Real-time PCR**. $AAV_9$-$mCaMKII\alpha$-$EGFP$-$P2A$-$iCre$ virus was injected into the mPFC of $Arrb2^{fl/fl}$ mice and their WT littermates. Four weeks later, the mPFC was dissected on ice immediately with Maxtrix for mouse (Plastic one Inc). Total RNA was extracted from mPFC of $Arrb2^{fl/fl}$ and WT mice using the TRIzol® Reagent (Thermo Fisher Scientific Inc, Waltham, MA, USA). The reverse transcription with random primers was conducted according to the Hiscript QRT SuperMix system (Nanjing Vazyme Biotech Co., Ltd) in one cycle program consisting of 2 min at 42 °C, 15 min at 50 °C, 2 min at 85 °C. Quantative PCR was performed in triplicates with ChamQ Universal SYBR qPCR Master Mix (Nanjing Vazyme Biotech Co., Ltd) by real-time PCR thermocycler (Eppendorf realplex2 Mastercycler ep realplex, Germany). The primers for CKO Arrb2 mRNA were designed flanking exon 2 (5'-GGGAGGGGAAGGAGGAGAAA-3' and 5'-TGCAGTTAGGGCTCGACTTC-3'). The primers for WT Arrb2 mRNA were designed within exon 2 (5'-GGGAG GGGAAGGAGGAGAAA-3' and 5'-TGCAGTTAGGGCTCGACTTC-3'). The primers for GAPDH are 5'-ACCACAGTCCATGCCATCAC-3' and 5'-TCCACC ACCCTGTTGCTGTA-3'.

**Cell culture**. Mouse neuroblastoma (N2a) cells were kindly provided by Professor Pei Gang (Shanghai Institutes for Biological Sciences, Chinese Academy of Sciences). N2a cells were plated at 20,000 cells per well in 12-well plates. After 20 h, cells were transiently transfected with AAV-hSyn-DIO-rM3Darr-mCherry/CAG-cre and β-arrestin2-shRNA with Lipofectamine 2000 Transfection Reagent (Invitrogen, Thermo Fisher Scientific). About 48 h later, cells were incubated overnight in serum-free medium before CNO (1 μM) stimulation. Then cells were harvested by Ripa lysis buffer (Beyotime, #P0013B) on ice.

**Western blotting**. The experiment mice were group housed with juvenile mice for 3 days. During sociability tests, the mice were exposed to the three-chamber containing a wire cage with a familiar or a novel mouse. For the control of sociability tests, the mice were stayed at homecage with or without optical stimulation. The mice were sacrificed 15 min after sociability test with or without optical stimulation. A standard three-chamber SRM task was performed in another cohort of mice that was sacrificed 15 min after sociability test followed by laser stimulation. The brains were removed immediately, and tissue samples from the mPFC or cell protein samples were homogenized with lysis buffer containing 1.0 μM phenylmethylsulfonyl fluoride (Beyotime, #P0013B), incubated on ice for 30 min, and centrifuged at 12,000 × g for 15 min at 4 °C. The protein concentration was determined by BCA assay (Thermo Fisher Scientific, #23227). Each sample was adjusted to a final protein concentration of 2 μg/μl, mixed with 6× SDS loading buffer (Beyotime, #P0015F), and boiled for 10 min at 95 °C. Samples were loaded onto 10% sodium dodecyl sulfate polyacrylamide gel electrophoresis (SDS-PAGE). Proteins were electrophoretically transferred from gels to nitrocellulose (NC) membranes (Whatman) that were then incubated with the following primary antibodies: anti-pERK[63] (1:1000 dilution, #9101 S, Cell Signaling Technology), anti-ERK[63] (1:2000, #4696 S, Cell Signaling Technology) at 4 °C overnight. The membranes were rinsed in Tris-buffered saline with 0.1% Tween-20 (TBST) 15 min for three times and then incubated with IRDye 700DX or 800DX-conjugated anti-rabbit or anti-mouse immunoglobulin G (IgG) (1:50,000; Rockland Immunochemicals Inc.) for 1.5 h at room temperature. Protein bands were visualized using Odyssey (LI-COR Biosciences). The immunoblots were analyzed with Image J software. ERK phosphorylation level was calculated by normalizing the intensity of pERK to the total ERK expression.

**Statistics and reproducibility.** Experimental data were presented as the mean ± s.e.m. and plotted with GraphPad Prism. Data from behavioral tests were analyzed by two-tailed Student's *t* test, one-way analysis of variance (ANOVA), or two-way ANOVA with repeated measures followed by Bonferroni's post hoc test with sessions as a within-subjects factor and AAV as a between-subjects factor. The photometry recording was analyzed with two-tailed Student's *t* test or one-way ANOVA for fluorescence transient. Immunofluorescence data were analyzed by two-tailed Student's *t* test. Western blotting data were analyzed by two-way ANOVA. The non-normalized data were analyzed with Mann–Whitney *U*-test and Kruskal–Wallis one-way ANOVA on Ranks. Full statistical analyses corresponding to each data set are presented in Supplementary Data 1. All the experiments were independently replicated in 4–15 mice, as reported in the figure. When possible, data was collected using biological replicates (multi-brain slices per animal analyzed for in situ hybridization experiments). Our sample sizes were estimated based on previous experience and are similar to those generally employed in the field.

**Reporting summary**. Further information on research design is available in the Nature Research Reporting Summary linked to this article.

## Data availability
All data needed to evaluate the conclusions in the paper are present in the paper and/or the Supplementary Material. Supplementary Fig. 10 contains uncropped versions of all blots in the paper. Supplementary Data 2 contains source data values underlying Figs. 1d, f, j, l, n, o, 2c, f, i–j, 3c–d, f–h, 4c–f, h, 5a, b, e–f, h, j, and 6b, d.

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

## Acknowledgements

We thank Dr. Gang Pei (Shanghai Institutes for Biological Sciences, Chinese Academy of Sciences) for Arrb2^fl/fl mice and Dr. Ken-ichiro Nakajima for R165L mutant M3 construct. This research was supported by the Science Technology Innovation 2030 Project of China (2021ZD0203500 to F.W. and L.M., 2021ZD0202104 to X.L.), National Natural Science Foundation of China Grants (32171041 to X.L., 31930046 to L.M., 82021002 to L.M.), the CAMS Innovation Fund for Medical Sciences (2021-I2M-5-009 to L.M. and X.L.), the Shanghai Municipal Science and Technology Major Project (2018SHZDZX01 to L.M.), ZJ Lab and Shanghai Center for Brain Science and Brain-Inspired Technology, and Chinese Postdoctoral Science Foundation (2021M690684 to D.C.).

## Author contributions

D.C., X.L., and L.M. planned the experiments. D.C. and J.W. performed the behavioral tests. D.C. and E.Y. carried out the stereotaxic surgery and biochemistry experiments. D.C. contributed to the cell culture and western blotting experiments. X.F. performed the FISH assay. D.C., E.Y., J.W. and F.W. analyzed the data. D.C. and X.L. drafted the manuscript. X.L. and L.M. revised the manuscript.

## Competing interests

The authors declare no competing interests.
