## [Peer Review File · Communications Biology]

Reviewers' comments:

Reviewer #1 (Remarks to the Author):

The authors elegantly demonstrate that social memory consolidation require NE release in the PFC. NE release likely acts mostly through activation of β -adrenergic receptors and subsequent β -arrestin2 recruitment. The experiments are thorough and mostly well-designed. The figures are clear and easy to understand. These are novel and interesting results. I only have one major comments regarding the authors claim. Unfortunately, the manuscript suffers from very poor English writing and needs to be thoroughly rewritten. I provide some suggestions after my comments.

Major points:

1/ When stimulating NE fibers in KO mice (Fig. 4c-d), the claim: " β 1-AR selective knockout in the mPFC impaired SRM consolidation, which was not rescued by optogenetic activation of LC-mPFC NE projection (Fig. 4c)." and "The results above indicate that the regulation of LC-mPFC NE release...". However, they previously showed that stimulation of LC fiber in the PFC is able to potentiate SRM 3 days but not 24h after learning. If the authors want to maintain the claim that SRM regulation by the LC-mPFC NE projection requires β -AR and β -arrestin2, they need to repeat these experiment and test for SRM 3 days after learning. As it stands, the conclusions of these experiments are the same than the ones of the simple KO experiments presented before.

2/ The writing of the article is extremely poor and difficult to follow. I strongly recommend using a professional copy-editing service.

3/ L77 "NE projections right after the exposure to an unfamiliar mouse" NE projections right after exposure to a novel mouse (sociability test) during the first part of the three-chamber test. Throughout the manuscript, the authors need to clearly define the 3-chamber test and its two phases (test of sociability then test of SRM). For example, you could refer the sociability and the SRM phase of the test respectively. The use of sociability test, SRM task and 3-chamber test is extremely confusing unless these terms are clearly defined and properly use.

Minor points:

1/ Given the design of the 3-chamber test, it is unclear whether mice show preference for social novelty or just novelty. Indeed, one could argue that the mice only care that the previously empty cup now contains something (a novel mouse). Please mention this point in the discussion.

2/ The Carvedilol infusion experiment cannot discard the possibility that long-term memory (3 days after learning) doesn't depend on G protein biased β -AR activation.

3/ Consider using "novel" instead of "unfamiliar" throughout the manuscript.

4/ L15 "The roles of norepinephrine (NE) signaling in the circuitry of the locus coeruleus (LC) to mPFC in SRM storage are unknown." Why do you focus on NE in the PFC?

5/ L100 "By western blotting of pERK" explain why you looked at ERK. Overall, it's unclear for the reader what is the relationship between β -arrestin2 and pERK.

6/ L140 "Sociability test significantly increased pERK levels in the mPFC of WT littermates, but not β -arrestin2 mPFC knockout mice (Fig. 3g)." Did you look at pERK levels following interaction with a familiar mouse?

7/ L56 " β -AR activation also facilitates excitatory synaptic transmission in mPFC excitatory neurons." Synaptic transmission is between two neurons. Do you mean it facilitates the post-synaptic response

of excitatory synapses on pyramidal neurons?

8/ "L104 LC-mPFC NE projection might regulate SRM consolidation through adrenergic downstream signaling, such as ERK activation." Explain the link between adrenergic signaling and ERK activation.

9/ L210 to 223: Consider moving all this to the introduction.

Writing issues:

L14 "helpful" critical

L18 " β -ARs" explain abbreviation

L20 "had no suppression" failed to suppress

L25 "consolidation in the aged mice." consolidation in aged mice

L35 "Social recognition memory (SRM) helps to establish and maintain social relationship, which is essential" "Social recognition memory (SRM) is critical to establish and maintain social relationships, which are essential"

L37 "SRM is assessed by the ability to distinguish the novel from the familiar conspecifics that have been encountered previously" SRM is assessed by the ability to distinguish and prefer novel from familiar conspecifics.

L38 "As other forms of learning, social information is acquired and undergoes consolidation process that a labile trace is stabilized into the long-term memory." As other forms of learning, social information is acquired and undergoes consolidation where a labile trace is stabilized into the long-term memory.

L40 "However, the nature of social memory consolidation is unclear." The mechanisms supporting social memory consolidation are unclear.

L44 "The elevation of cellular excitation of excitatory neurons in the mPFC decreases social exploration" Activating excitatory neurons in the mPFC

L52 "... broadly project to forebrain through highly ramified axonal arborization, including the mPFC" broadly project to the forebrain, including the mPFC through highly ramified axonal arborization.

L54 "The patch-clamp recording shows that NE release enhances mPFC neuronal intrinsic excitability, which was prevented by blockade of β -adrenergic receptor (β -ARs)." Patch-clamp recording shows that NE release enhances mPFC neuronal intrinsic excitability, which was prevented by blockade of β -adrenergic receptor (β -ARs)."

L57 "However, there is a lack of knowledge whether the LC-mPFC NE system involves in and how the NE signaling in the mPFC contributes to the SRM consolidation." However, whether the LC-mPFC NE system is involved in SRM and how NE signaling in the mPFC could contribute to the SRM consolidation is unknown.

L80 "Social memory tests were performed 1 hour or 1 day or 3 days after the exposure to the unfamiliar mouse in sociability test by comparing preference for the novel mouse." The SRM test was performed 1 hour, 1 day or 3 days following exposure to the novel mouse by comparing interaction time with the now familiar mouse versus a second novel mouse.

L84 "suggesting a similar sociability in these mice." Suggesting sociability was unchanged.

L86 "Optogenetic inhibition of LC-mPFC NE projection 21 did not change preference for the novel mouse 1 in social memory test 1 h later (Fig.1c), but greatly decreased exploration for the novel mouse 2 in social memory test 1 day later (Fig. 1d)," rephrase

L108 "By high-resolution fluorescence in situ hybridization by RNAscope (FISH)," Using high-resolution fluorescence in situ hybridization...

L111, L113, 180, 181 and elsewhere. Remove capitals for "Novel Mice"

L115 "All the mice in the SRM tasks showed significant preference for the Unfamiliar Mouse than the empty cage in the sociability test" This is utterly confusing. If you described the results of the sociability test why talking about SRM? I would say "all mice preferred the novel mouse to the empty cage during the sociability test."

L122 "retain the social memory" display social memory

L159 "All the mice in the SRM tasks showed significant preference for the unfamiliar mouse than the empty cage in the sociability test (Supplementary Fig. 6)." See previous comments and suggestions.

L165 "To pharmacologically manipulate β -arrestin biased signaling pathway, the R165L mutant rM3Dq (rM3Darr) was introduced in SRM task to selectively activate β -arrestin dependent signaling by CNO treatment" rephrase

L176 "was detected" remove

L194 "Our results showed that SRM consolidation was impaired in aged mice" Our results show that SRM consolidation is impaired in aged mice.

L196 and elsewhere "of the aged mice" of aged mice

Reviewer #2 (Remarks to the Author):

Based on the previous finding that β -AR/ β -arrestin signaling in the entorhinal cortex mediates the consolidation of object recognition memory (ref 19), this study sought to prove that β -AR/ β -arrestin signaling in the prefrontal cortex mediates the consolidation of social recognition memory (SRM). While the conceptual novelty is modest, multiple cutting-edge approaches, including optogenetic, chemogenetic, conditional knockout, and pharmacological, have been used in this study. Experiments are well designed and data are comprehensive. There are a few concerns that need to be addressed to strengthen the conclusions.

1. Many places lack the critical evidence about the manipulation. For example, it states "we selectively knocked out β -arrestin2 in the mPFC (Fig. 3e, f)" (p7, line 137-138), however, it lacks direct evidence that β -arrestin2 is indeed knocked out. The same is true for Adrb1 knockout or Adrb2 knockout.

2. Fig. 1j and 1k, manipulating LC-mPFC NE pathway changes p-ERK, but it does not prove that ERK activation is the downstream mechanism for adrenergic regulation of SRM.

3. Where is the evidence that Carvedilol is the "G protein biased β -AR antagonist" (p7, line 130) or "the β -blocker that can mildly activate β -arrestin-biased signaling" (Abstract)?

4. What is the difference between animals used in Fig. 3h and Fig. 4g, since both are with "selective deletion of β -arrestin2 in the mPFC" (p7, line 137-138; p8, line 156)?

5. Using pERK to confirm the activation of β -arrestin signaling by rM3Darr (Fig. 5a) is not compelling, since pERK can be activated by many GPCR signaling. It needs more specific readout for β -arrestin activation.

6. Many prior studies have revealed the neurotransmitters, neuropeptides and circuits involved in SRM (ref. 25-31). The current study points to β -AR/ β -arrestin signaling in LC to PFC circuit for SRM. Is there any connection among these different findings?

7. It is puzzling why SRM is dependent on β -arrestin signaling instead of GPCR signaling downstream of β -AR. What could uniquely link β -arrestin to SRM?

8. The same diagrams for "Habituation", "Sociability test" and "Social memory test" are shown repeatedly in all the figures (11 times in total), which is completely unnecessary, and should be removed. Only the unique parts can be kept.

9. Throughout the paper, "biased" was used repeatedly for a large number of times (6 in Abstract alone), many of which should be replaced with a better wording or simply changed to " β -arrestin signaling".

10. Page 6, line 110-114, several references for corresponding figures are incorrect (e.g. Fig. 2 b-d should be changed to Fig. 2c-e, Fig. 2e should be changed to Fig. 2f, etc).

11. Why the optical stimulation of ChrimsonR+ NE terminals (5 mW, 5 ms pulse, 1 s duration) vs. eNpHR3.0 terminals (10 mW, 10 s duration) in the mPFC (p20) are very different? Are they all trains of 5-ms pulses with 1 s or 10 s duration?

Reviewer #3 (Remarks to the Author):

The manuscript by Cheng et al. examines the role of adrenergic signaling by locus coeruleus (LC)-derived norepinephrine (NE) in the medial prefrontal cortex (mPFC) during the consolidation of social recognition memory (SRM). Through both genetic and pharmacologic approaches directed to the LC and mPFC, they find that beta-adrenergic signaling is required for SRM consolidation. Using similar approaches, they also implicate beta-arrestin2 signaling as being required downstream of beta-adrenergic signaling. Finally, the authors demonstrate that enhancing beta-arrestin signaling in the mPFC of aged mice during the consolidation period restores SRM that is normally impaired by age. In general the experiments are clearly reported, utilize informative techniques, and their results generally support the conclusions that the authors provide. Listed below are items that, if addressed, would enhance the manuscript and its studies.

1) Results, page 5, lines 77 & 85: the authors state that they are stimulating LC NE terminals when in fact they are attempting to inhibit the activity of those terminals.

2) Results, page 5, lines 81 & 95; Fig 1 legend, line 546; Fig 5 legend, line 613: in all these cases it is written that memory testing was performed 3 days later, when in fact it was performed 4 days later (i.e. after training, equivalent to 3 days after the 1 day memory test).

3) Results, page 5, line 88: the authors state "greatly decreased exploration of the novel mouse", which is incorrect. The manipulation increased exploration of the familiar mouse (and decreased the discrimination index).

4) Results, page 6, line 109: the authors state "greater density than beta2-ARs". It is unlikely that the two different probes can be compared quantitatively. How does one know that the beta2 probe gives a signal that is equivalent to the beta1 probe?

5) Results, page 6, line 110: RNAscope should be used to document the extent of floxed deletion in the mPFC for all targets: beta1, beta2, and beta-arrestin2.

6) Results, page 7: there's no mention of performing experiments to examine the role of alpha1-ARs, despite this being mentioned in the Discussion.

7) Results, page 7, line 130: how do we know that carvedilol didn't have an effect simply because the dose used is too low?

8) Results, page 7, line 138: why was beta-arrestin2 studied and not also beta-arrestin1?

9) Results, page 8, lines 156 & 157: it is not at all clear how expression of Flp recombinase in the LC area leads to expression of Cre in mPFC neurons, since these are different cellular compartments. Further, AAV-hSyn-FlpO is not selective for adrenergic neurons, so transfected neurons are likely adrenergic and non-adrenergic.

10) Discussion: Since it appears that beta1-AR is most relevant, and beta-arrestin2 is also quite

relevant and likely downstream of beta1-AR, the general literature on beta-arrestin2-mediated signaling by beta1-AR should be discussed.

11) Methods, page 15, lines 291-3: the authors state that the mice are on a reversed 12 h light/dark cycle, which usually means that the light cycle is during the investigators' night time. Subsequently the authors state that experiments were performed during the light cycle. Were all experiments truly performed during the investigators' night time?

12) Methods, page 15, line 305: Sigma P0884 is (+/-)-propranolol, not (+)-propranolol.

13) Methods, page 16: only some of the viral vectors used are listed here, while all should be listed.

14) Fig 5, panel h: AAV-CAG-Cre is missing from the vector list above the brain section outline.

15) Supplemental figures: For the open field data, instead of "distance in the central area", it would be more informative to present % distance in the central area since the total distance of each animal varies. Also, the open field and the 0 maze present data in a manner such that quantities decrease with increased anxiety. It would be nice to be consistent with this and present the light-dark box data as time in the light box (rather than the black box).

Point-by-point response to Reviewer #1:

Reviewer #1 (Remarks to the Author):

The authors elegantly demonstrate that social memory consolidation require NE release in the PFC. NE release likely acts mostly through activation of β -adrenergic receptors and subsequent β -arrestin2 recruitment. The experiments are thorough and mostly well-designed. The figures are clear and easy to understand. These are novel and interesting results. I only have one major comments regarding the authors claim. Unfortunately, the manuscript suffers from very poor English writing and needs to be thoroughly rewritten. I provide some suggestions after my comments.

Major points:

1) *When stimulating NE fibers in KO mice (Fig. 4c-d), the claim: “ β 1-AR selective knockout in the mPFC impaired SRM consolidation, which was not rescued by optogenetic activation of LC-mPFC NE projection (Fig. 4c).” and “The results above indicate that the regulation of LC-mPFC NE release...”. However, they previously showed that stimulation of LC fiber in the PFC is able to potentiate SRM 3 days but not 24h after learning. If the authors want to maintain the claim that SRM regulation by the LC-mPFC NE projection requires β -AR and β -arrestin2, they need to repeat these experiment and test for SRM 3 days after learning. As it stands, the conclusions of these experiments are the same than the ones of the simple KO experiments presented before.*

We thank the reviewer for the important suggestion. We tested SRM 4 days later in Fig. 4c, but we did not include them in the previous manuscript. Likewise, we have added these results as suggested. The results showed that the WT mice with activation of LC-mPFC NE projections showed persistently greater preference for the novel mouse than that of the control group, and the mice with deletion of β 1-AR or β -arrestin2 in the mPFC showed impaired discrimination of the novel mouse 4 days after training even with activation of LC-mPFC NE projections (Fig. 4d, f, Page 9, Line 173-178).

2) *The writing of the article is extremely poor and difficult to follow. I strongly recommend using a professional copy-editing service.*

As suggested, we have sent our manuscript for main text editing with Nature Research Editing Service.

3) *L77 “NE projections right after the exposure to an unfamiliar mouse” NE projections right after exposure to a novel mouse (sociability test) during the first part of the three-chamber test. Throughout the manuscript, the authors need to clearly define the 3-chamber test and its two phases (test of sociability then test of SRM). For example, you could refer the sociability and the SRM phase of the test respectively. The use of sociability test, SRM task and 3-chamber test is extremely confusing unless these terms are clearly defined and properly use.*

As suggested, we have added information of three-chamber social recognition memory task in the methods and results, and have defined sociability test and SRM test respectively (Page 5-6, Line 81-107; Page 19-20, Line 394-425).

Minor points:

1) Given the design of the 3-chamber test, it is unclear whether mice show preference for social novelty or just novelty. Indeed, one could argue that the mice only care that the previously empty cup now contains something (a novel mouse). Please mention this point in the discussion.

The three-chamber task is a widely used test paradigm to quantitatively measure sociability and social memory (Zimprich A, et al., Curr Protoc Mouse Biol. 2017). The mouse has a real choice to enter one of the three chambers, thus measurement of sociability is allowed by direct comparison of the preference for a wired cage with a novel mouse over an empty cage. With this task, the measurements of social recognition memory are also permitted by comparing the interaction with a novel mouse over the familiar mice in each chamber. However, it is still unclear whether mice show preference for social novelty or just novelty in the wired cage. It is possible that the mice might only care that the previously empty cage now contains something or a novel mouse. We have added this in the discussion (Page 11-12, Line 230-238).

2) The Carvedilol infusion experiment cannot discard the possibility that long-term memory (3 days after learning) doesn't depend on G protein biased β -AR activation.

Our results showed that carvedilol did not suppress social memory in SRM Test 1 as propranolol did, suggesting that SRM one day after training might not depend on G protein biased β -AR signaling. As suggested, we cannot exclude the possibility that SRM 4 days after training does not depend on G protein biased β -AR activation. To examine the role of carvedilol on SRM maintenance, we have examined the effects of carvedilol on SRM tested 4 days after training. The mice with treatment of carvedilol did not show a further increase of preference for the novel mouse in social memory test 1 day later, but showed a persistent preference for the novel mouse tested 4 days later. Thus, activation of β -AR/ β -arrestin pathway might prolong social memory maintenance and promote SRM consolidation.

Fig. Carvedilol promoted SRM maintenance.

a Experimental scheme. β -AR antagonist, carvedilol (3 mg/kg, i.p) was infused immediately after sociability test and social memory tests were carried out 1 day or 4 days later. **b, c** Statistical graphs of exploration time and discrimination scores [Vehicle: $n = 13$, Carvedilol: $n = 13$. **b**, Left: $F_{\text{mouse} \times \text{treatment}}(1, 24) = 1.079, p = 0.309$, two-way RM ANOVA; Right: $t(24) = -1.370, p = 0.183$, two-tailed Student's t -test. **c**, Left: $F_{\text{mouse} \times \text{treatment}}(1, 24) = 11.775, p = 0.002$, two-way RM ANOVA; Right: $t(24) = -3.498, p = 0.002$, two-tailed Student's t -test]. ** $p < 0.01$ and *** $p < 0.001$ vs indicated group.

3) Consider using “novel” instead of “unfamiliar” throughout the manuscript.

We have replaced “unfamiliar” with “novel” throughout the manuscript (Page 5, Line 90; Page 6, Line 105; Page 7, Line 124; Page 8, Line 146; Page 9, Line 175; Page 10, Line 204, Line 210; Page 19, Line 403).

4) L15 “The roles of norepinephrine (NE) signaling in the circuitry of the locus coeruleus (LC) to mPFC in SRM storage are unknown.” Why do you focus on NE in the PFC?

The medial prefrontal cortex (mPFC) is a critical brain area for SRM storage. LC-mPFC NE can regulate mPFC neuronal intrinsic excitability and facilitate post-excitatory synaptic transmission, however, the roles of NE signaling in the circuitry of the locus coeruleus (LC) to mPFC in SRM storage are unknown. We have added this in the abstract (Page 2, Line 16-20).

5) L100 “By western blotting of pERK” explain why you looked at ERK. Overall, it’s unclear for the reader what is the relationship between β -arrestin2 and pERK.

The activation of β -adrenergic receptors recruits ERK to facilitate long-term potentiation maintenance and long-term memory formation (Gelinias JN, et al. JBC.

2007; Liu X, et al., Proc Natl Acad Sci U S A. 2015). Thus, we examined pERK levels in the mPFC after sociability test with laser stimulation.

β -Arrestins, which are downstream of GPCRs, act as signal transducers and mediate the activation of a diverse array of signaling and cellular responses, including ERK activation (Kim IM, et al., Proc Natl Acad Sci U S A. 2008; Shenoy SK, et al., JBC. 2006; Ahn S, et al., JBC. 2004).

We have added the information in the results (Page 6, Line 107-110; Page 8, Line 157-159)

6) L140 *“Sociability test significantly increased pERK levels in the mPFC of WT littermates, but not β -arrestin2 mPFC knockout mice (Fig. 3g).” Did you look at pERK levels following interaction with a familiar mouse?*

As suggested, we have added a new experiment and examined pERK levels in the mPFC after interaction with a familiar or a novel mouse alone. All the experiment mice were group housed with juvenile mice for three days. During sociability tests, the mice were exposed to the three-chamber containing a wire cage with a familiar or a novel mouse. The results showed that exposure to a novel mouse significantly increased pERK levels in the mPFC compared to the mice with exposure to a familiar mouse or the mice in control group (Supplementary Fig. 2, Page 6, Line 110-113).

7) L56 *“ β -AR activation also facilitates excitatory synaptic transmission in mPFC excitatory neurons.” Synaptic transmission is between two neurons. Do you mean it facilitates the post-synaptic response of excitatory synapses on pyramidal neurons?*

We have rephrased this sentence. “ β -AR activation also facilitates post-synaptic response of excitatory synapses on pyramidal neurons in the mPFC” (Page 4, Line 61-62).

8) *“L104 LC-mPFC NE projection might regulate SRM consolidation through adrenergic downstream signaling, such as ERK activation.” Explain the link between adrenergic signaling and ERK activation.*

Thank you for the suggestion. The activation of β -adrenergic receptors recruits ERK to facilitate long-term potentiation maintenance and long-term memory formation (Gelines JN, et al. JBC. 2007; Liu X, et al., Proc Natl Acad Sci U S A. 2015). Thus, we examined pERK levels in the mPFC after sociability test with laser stimulation. The information has been added (Page 6, Line 107-110).

9/ L210 to 223: *Consider moving all this to the introduction.*

We have moved this part to the introduction and rewritten the introduction (Page 3, Line 44-55).

Writing issues:

L14 *“helpful” critical*

L18 *“ β -ARs” explain abbreviation*

L20 *“had no suppression” failed to suppress*

L25 *“consolidation in the aged mice.” consolidation in aged mice*

L35 *“Social recognition memory (SRM) helps to establish and maintain social relationship, which is essential” “Social recognition memory (SRM) is critical to establish and maintain social relationships, which are essential”*

L37 *“SRM is assessed by the ability to distinguish the novel from the familiar conspecifics that have been encountered previously” SRM is assessed by the ability to distinguish and prefer novel from familiar conspecifics.*

L38 *“As other forms of learning, social information is acquired and undergoes consolidation process that a labile trace is stabilized into the long-term memory.” As other forms of learning, social information is acquired and undergoes consolidation where a labile trace is stabilized into the long-term memory.*

L40 *“However, the nature of social memory consolidation is unclear.” The mechanisms supporting social memory consolidation are unclear.*

L44 *“The elevation of cellular excitation of excitatory neurons in the mPFC decreases social exploration” Activating excitatory neurons in the mPFC*

L52 *“... broadly project to forebrain through highly ramified axonal arborization, including the mPFC” broadly project to the forebrain, including the mPFC through highly ramified axonal arborization.*

L54 *“The patch-clamp recording shows that NE release enhances mPFC neuronal intrinsic excitability, which was prevented by blockade of β -adrenergic receptor (β -ARs).” Patch-clamp recording shows that NE release enhances mPFC neuronal intrinsic excitability, which was prevented by blockade of β -adrenergic receptor (β -ARs).”*

L57 *“However, there is a lack of knowledge whether the LC-mPFC NE system involves in and how the NE signaling in the mPFC contributes to the SRM consolidation.” However, whether the LC-mPFC NE system is involved in SRM and how NE signaling in the mPFC could contribute to the SRM consolidation is unknown.*

L80 *“Social memory tests were performed 1 hour or 1 day or 3 days after the exposure to the unfamiliar mouse in sociability test by comparing preference for the novel mouse.” The SRM test was performed 1 hour, 1 day or 3 days following exposure to the novel mouse by comparing interaction time with the now familiar mouse versus a second novel mouse.*

L84 *“suggesting a similar sociability in these mice.” Suggesting sociability was unchanged.*

L86 *“Optogenetic inhibition of LC-mPFC NE projections²¹ did not change preference for the novel mouse 1 in social memory test 1 h later (Fig. 1c), but greatly decreased exploration for the novel mouse 2 in social memory test 1 day later (Fig. 1d),” rephrase*

L108 *“By high-resolution fluorescence in situ hybridization by RNAscope (FISH),” Using high-resolution fluorescence in situ hybridization...*

L111, L113, 180, 181 and elsewhere. *Remove capitals for “Novel Mice”*

L115 *“All the mice in the SRM tasks showed significant preference for the Unfamiliar Mouse than the empty cage in the sociability test” This is utterly confusing. If you described the results of the sociability test why talking about SRM? I would say “all mice preferred the novel mouse to the empty cage during the sociability test.”*

L122 “retain the social memory” display social memory

L159 “All the mice in the SRM tasks showed significant preference for the unfamiliar mouse than the empty cage in the sociability test (Supplementary Fig. 6).” See previous comments and suggestions.

L165 “To pharmacologically manipulate β -arrestin biased signaling pathway, the R165L mutant rM3Dq (rM3Darr) was introduced in SRM task to selectively activate β -arrestin dependent signaling by CNO treatment” rephrase

L176 “was detected” remove

L194 “Our results showed that SRM consolidation was impaired in aged mice” Our results show that SRM consolidation is impaired in aged mice.

L196 and elsewhere “of the aged mice” of aged mice.

The writing has been rephrased as suggested above (Line 15, 22, 29, 37, 39, 40, 41-42,51, 57-58, 59, 62, 93-95, 95-98, 121,124, 127, 203, 204, 127, 135, 145-147, 187-189, 199, 216-217, 218).

Reviewer #2 (Remarks to the Author):

Based on the previous finding that β -AR/ β -arrestin signaling in the entorhinal cortex mediates the consolidation of object recognition memory (ref 19), this study sought to prove that β -AR/ β -arrestin signaling in the prefrontal cortex mediates the consolidation of social recognition memory (SRM). While the conceptual novelty is modest, multiple cutting-edge approaches, including optogenetic, chemogenetic, conditional knockout, and pharmacological, have been used in this study. Experiments are well designed and data are comprehensive. There are a few concerns that need to be addressed to strengthen the conclusions.

*1. Many places lack the critical evidence about the manipulation. For example, it states “we selectively knocked out β -arrestin2 in the mPFC (Fig. 3e, f)” (p7, line 137-138), however, it lacks direct evidence that β -arrestin2 is indeed knocked out. The same is true for *Adrb1* knockout or *Adrb2* knockout.*

Thank you for the important suggestion. Knockout efficiency of *Adrb1* and *Adrb2* has been verified by multiplexed single-molecule RNA fluorescence in situ hybridization (smFISH) with RNAscope (Fig. 2a-d). Regard to the strategy of *Arrb2^{fl/fl}* mouse development, the loxP fragments are inserted into the introns besides the exon2 that can be deleted by Cre recombinase. However, the exon2 (31bps) is too short for RNAscope. So, we have to apply qPCR to check knockout efficiency of β -arrestin2 in *Arrb2^{fl/fl}* mice with injection of *AAV-mCaMKII α -eGFP-P2A-iCre* in the mPFC (Fig. 3f). The results have been added (Fig. 2a-d and 3f, Page 6-7 and 8, Line 121-123 and 151-153).

2. Fig. 1j and 1k, manipulating LC-mPFC NE pathway changes p-ERK, but it does not prove that ERK activation is the downstream mechanism for adrenergic regulation of SRM.

The activation of β -adrenergic receptors recruits ERK to facilitate long-term potentiation maintenance and long-term memory formation (Gelinas JN, et al. JBC. 2007; Liu X, et al., Proc Natl Acad Sci U S A. 2015). Thus, we examined pERK levels in the mPFC after sociability test with laser stimulation. We have added the information in the results (Page 6, Line 107-110).

3. Where is the evidence that Carvedilol is the “G protein biased β -AR antagonist” (p7, line 130) or “the β -blocker that can mildly activate β -arrestin-biased signaling” (Abstract)?

Thank you for the reminder. The references of carvedilol have been added (Page 7, Line 144).

4. What is the difference between animals used in Fig. 3h and Fig. 4g, since both are with “selective deletion of β -arrestin2 in the mPFC” (p7, line 137-138; p8, line 156)?

Thank you for the reminder. In Fig. 3h, we injected *AAV-mCaMKII α -eGFP-P2A-iCre* in the mPFC of *Arrb2^{fl/fl}* mice and selectively knocked out β -arrestin2 in mPFC

excitatory neurons. In Fig. 4g, we injected anterograde *scAAV1-hSyn-FlpO* in the LC and *AAV-EF1 α -fDIO-Cre-mCherry* in the mPFC of *Arrb2^{fl/fl}* mice, which allowed expression of Cre recombinase and then knockout of β -arrestin2 in mPFC neurons that were innervated by LC. We have added the information in the results (Page 8, Line 153-155; Page 9, Line 178-181).

5. *Using pERK to confirm the activation of β -arrestin signaling by rM3Darr (Fig. 5a) is not compelling, since pERK can be activated by many GPCR signaling. It needs more specific readout for β -arrestin activation.*

Upon ligand binding, GPCRs undergo conformational changes that promote their binding to heterotrimeric G proteins and β -arrestins. Although both G protein and β -arrestins mediate ERK activation, pERK is often used for specifying β -arrestin biased signaling pathway when G protein pathway is blocked (Wang Y, et al., *Am J Physiol Renal Physiol.* 2017; Erickson CE, et al., *PLoS One.* 2013; Wang J, et al., *Nat Commun.* 2017).

rM3Darr is a M3 muscarinic receptor-based DREADD containing a point mutation within the highly conserved DRY motif [Rq(R165L)] that lacks the ability to activate heterotrimeric G proteins, but retains the intact ability to recruit β -arrestin in a CNO dependent fashion (Nakajima K, et al. *Mol Pharmacol.* 2012). Dr. Ken-ichiro Nakajima found that rM3Darr was unable to couple to Gq and increase the second message, such as cAMP, but could dose-dependently increase β -arrestin dependent ERK activation. Rq(R165L) M3 construct is the gift from Dr. Ken-ichiro Nakajima. We subcloned rM3Darr into an AAV plasmid and verified the results from Dr. Ken-ichiro Nakajima that rM3Darr was able to promote ERK phosphorylation in β -arrestin and CNO-dependent fashion.

6. *Many prior studies have revealed the neurotransmitters, neuropeptides and circuits involved in SRM (ref. 25-31). The current study points to β -AR/ β -arrestin signaling in LC to PFC circuit for SRM. Is there any connection among these different findings?*

We have moved this part from discussion to introduction and rearranged it (Page 3, Line 44-47).

7. *It is puzzling why SRM is dependent on β -arrestin signaling instead of GPCR signaling downstream of β -AR. What could uniquely link β -arrestin to SRM?*

Upon ligand binding, G protein-coupled receptors (GPCRs), including β -ARs, undergo conformational changes that promote their binding to heterotrimeric G proteins and β -arrestins. Recent studies indicate that both heterotrimeric G proteins and β -arrestins are capable of interacting with and recruiting intracellular signaling molecules independently (Reiter E, et al., *Annu Rev Pharmacol Toxicol.* 2012; Rajagopal S, et al.,

Nat Rev Drug Discov. 2010). Most ligands that bind to GPCRs have balanced or unbiased activity for signaling through G proteins and β -arrestins. Some ligands displaying bias towards one pathway over the other pathways have also been discovered. Carvedilol is the G protein biased β -AR antagonist with mild activation of β -arrestin

signaling. Propranolol is a full antagonist that blocks both G protein and β -arrestin pathways. Previous studies have shown that carvedilol selectively promotes the recruitment of G α i to β 1-AR, activating the β -arrestin2 biased pathway, and inducing β 1-AR-mediated transactivation of the EGFR and β -arrestin2-dependent ERK activation, suggesting that β -arrestin2 can mediate β 1-AR signaling independently. Our results showed that carvedilol did not suppress SRM in memory retention test 1 as propranolol did, suggesting that SRM consolidation after training might not depend on G protein biased pathway. Furthermore, the mice with β -arrestin 2 knockout in the mPFC showed impaired SRM. When β -arrestin biased signaling pathway was activated after training, SRM maintenance was promoted and the impaired SRM by LC-mPFC projection inhibition or β 1-AR deletion was rescued. These results suggest that SRM consolidation is dependent on β -arrestin-biased β -AR signaling pathway in the mPFC. We have added this in the discussion (Page 13-14, Line 253-265, Line 275-281).

8. *The same diagrams for “Habituation”, “Sociability test” and “Social memory test” are shown repeatedly in all the figures (11 times in total), which is completely unnecessary, and should be removed. Only the unique parts can be kept.*
As suggested, we have removed unnecessary diagrams in the figures.

9. *Throughout the paper, “biased” was used repeatedly for a large number of times (6 in Abstract alone), many of which should be replaced with a better wording or simply changed to “ β -arrestin signaling”.*
As suggested, we have removed unnecessary “biased” in the manuscript.

10. *Page 6, line 110-114, several references for corresponding figures are incorrect (e.g. Fig. 2 b-d should be changed to Fig. 2c-e, Fig. 2e should be changed to Fig. 2f, etc).*
Thank you for the reminder. The references for Fig. 2 have been corrected (Page 7, Line 123, 125, 126).

11. *Why the optical stimulation of ChrimsonR+ NE terminals (5 mW, 5 ms pulse, 1 s duration) vs. eNpHR3.0 terminals (10 mW, 10 s duration) in the mPFC (p20) are very different? Are they all trains of 5-ms pulses with 1 s or 10 s duration?*
Different opsins have distinct ways of stimulation. To examine whether optical stimulation can promote or inhibit NE release from LC-mPFC terminals, we did in vivo recording of NE release in the mPFC with a brief stimulation protocol (5 mW, twenty 5-ms pulses at 5, 25, and 50 Hz in 1 s duration for ChrimsonR⁺ NE terminals and 10 mW, 10 s duration for eNpHR3.0 terminals). In the behavioral experiments, we applied optical stimulation according to previous studies (Takeuchi, T. et al., Nature. 2016; Klapoetke NC, et al. Nat Methods. 2014). To examine the effects of activation of NE terminals (ChR2⁺) on SRM, optical stimulation (473 nm, 5 mW, twenty 5-ms pulses at 25 Hz, every 5 s for the duration of 5 min) was delivered. To examine the effects of inhibition of NE terminals (eNpHR3.0⁺) on SRM, optical stimulation (590 nm, 10 mW, constantly for 10 min) was delivered.

Reviewer #3 (Remarks to the Author):

The manuscript by Cheng et al. examines the role of adrenergic signaling by locus coeruleus (LC)-derived norepinephrine (NE) in the medial prefrontal cortex (mPFC) during the consolidation of social recognition memory (SRM). Through both genetic and pharmacologic approaches directed to the LC and mPFC, they find that beta-adrenergic signaling is required for SRM consolidation. Using similar approaches, they also implicate beta-arrestin2 signaling as being required downstream of beta-adrenergic signaling. Finally, the authors demonstrate that enhancing beta-arrestin signaling in the mPFC of aged mice during the consolidation period restores SRM that is normally impaired by age. In general the experiments are clearly reported, utilize informative techniques, and their results generally support the conclusions that the authors provide. Listed below are items that, if addressed, would enhance the manuscript and its studies.

1) Results, page 5, lines 77 & 85: the authors state that they are stimulating LC NE terminals when in fact they are attempting to inhibit the activity of those terminals.

We are sorry for the confusing description. We injected AAV-EF1 α -DIO-hChR2(H134R)-mCherry or AAV-EF1 α -DIO-eNpHR3.0-EYFP in the LC of TH-Cre mice and expressed eNpHR3.0-EYFP or ChR2-mCherry in LC NE neurons. We aimed to activate LC NE terminals by stimulating ChR2 and inhibit LC NE terminals by stimulating eNpHR3.0. We have rephrased this part in the results (Page 5, Line 82-87).

2) Results, page 5, lines 81 & 95; Fig 1 legend, line 546; Fig 5 legend, line 613: in all these cases it is written that memory testing was performed 3 days later, when in fact it was performed 4 days later (i.e. after training, equivalent to 3 days after the 1 day memory test).

Thank you for the suggestion. We have corrected this accordingly (Page 6, Line 103; Page 9, Line 177; Page 28, Line 587, Page 32, Line 654).

3) Results, page 5, line 88: the authors state “greatly decreased exploration of the novel mouse”, which is incorrect. The manipulation increased exploration of the familiar mouse (and decreased the discrimination index).

Thank you for pointing out this incorrect description. We have corrected this accordingly (Page 5, Line 97-98).

4) Results, page 6, line 109: the authors state “greater density than beta2-ARs”. It is unlikely that the two different probes can be compared quantitatively. How does one know that the beta2 probe gives a signal that is equivalent to the beta1 probe?

Thank you for the suggestion. We have deleted the comparison (Page 6, Line 121).

5) Results, page 6, line 110: RNAscope should be used to document the extent of floxed deletion in the mPFC for all targets: beta1, beta2, and beta-arrestin2.

As suggested, we have checked knockout efficiency of β 1-AR and β 2-AR in *Adrb1^{fl/fl}* and *Adrb2^{fl/fl}* mice after injection of *AAV-mCaMKII α -eGFP-P2A-iCre* in the mPFC with RNAscope (Fig. 2a-d).

Regarding the strategy of *Arrb2^{fl/fl}* mouse development, the loxP fragments are inserted into the introns besides the exon2 that can be deleted by Cre recombinase. The exon2 (31bps) is too short for RNAscope. So, we have to apply RT-PCR to check knockout efficiency of β -arrestin2 in *Arrb2^{fl/fl}* mice with injection of *AAV-mCaMKII α -eGFP-P2A-iCre* in the mPFC (Fig. 3f).

The results have been added (Fig. 2a-d and 3f, Page 6-7 and 8, Line 121-123 and 151-153).

6) *Results, page 7: there's no mention of performing experiments to examine the role of alpha1-ARs, despite this being mentioned in the Discussion.*

Thank you for the suggestion. Carvedilol is the non-selective β -AR G protein biased antagonist and α 1-AR antagonist. While no changes of SRM were detected by administration of carvedilol in the mPFC, the roles of α 1-AR in social memory consolidation still deserve investigation. We do not have α 1-AR CKO mice in our lab now. We will consider the relevant experiments in future investigations.

7) *Results, page 7, line 130: how do we know that carvedilol didn't have an effect simply because the dose used is too low?*

In our previous study, we found carvedilol (10 μ g, i.c.v.) was able to increase pERK levels in the entorhinal cortex (Liu X, et al., P Proc Natl Acad Sci U S A. 2015). We suppose that carvedilol (5 μ g) infused in the mPFC is enough, and we hope to avoid overdose when injected locally.

8) *Results, page 7, line 138: why was beta-arrestin2 studied and not also beta-arrestin1?*

In our previous studies, we found β -arrestin1 and β -arrestin2 have different roles in memory storage. The β -arrestin1 KO mice did not show impairment of object recognition memory reconsolidation (Liu X, et al., Proc Natl Acad Sci U S A. 2015) or fear memory consolidation (data not shown in Li YT, et al., Proc Natl Acad Sci U S A. 2009). One recent study shows that β -arrestin1 regulates lactate metabolism and contributes to β 2-AR dependent memory formation (Liu CH, et al., Biol Psychiatry. 2017). The role of β -arrestin1 in SRM needs future investigation.

9) *Results, page 8, lines 156 & 157: it is not at all clear how expression of Flp recombinase in the LC area leads to expression of Cre in mPFC neurons, since these are different cellular compartments. Further, AAV-hSyn-FlpO is not selective for adrenergic neurons, so transfected neurons are likely adrenergic and non-adrenergic.*
Thank you for the reminder. We applied anterograde transsynaptic viral vector in Fig. 4g. We injected anterograde *scAAV1-hSyn-FlpO* in the LC that allowed FlpO transsynaptic expression in the downstream areas of the LC, and injected *AAV-EF1 α -fDIO-Cre-mCherry* in the mPFC of *Arrb2^{fl/fl}* mice, which allowed expression of Cre

recombinase and then knockout of β -arrestin2 in mPFC neurons that innervated by the LC. We have added the information in the results (Page 9, Line 178-181).

10) Discussion: Since it appears that beta1-AR is most relevant, and beta-arrestin2 is also quite relevant and likely downstream of beta1-AR, the general literature on beta-arrestin2-mediated signaling by beta1-AR should be discussed.

Thank you for the suggestion. Studies show that carvedilol selectively promotes recruitment of G α i to β 1-AR to activate β -arrestin2 biased pathway (Wang J, et al., Nat Commun. 2017; Wang J, et al., Mol Pharmacol. 2021), and induces β 1-AR mediated transactivation of the EGFR and β -arrestin2 dependent ERK activation (Kim IM, et al., Proc Natl Acad Sci U S A. 2008), suggesting β -arrestin2 is able to mediate β 1-AR signaling independently. We have added these in the discussion (Page 13, Line 261-265).

11) Methods, page 15, lines 291-3: the authors state that the mice are on a reversed 12 h light/dark cycle, which usually means that the light cycle is during the investigators' night time. Subsequently the authors state that experiments were performed during the light cycle. Were all experiments truly performed during the investigators' night time? Mice are nocturnal rodents displaying peak performance in learning during the dark phase when they are active (Nelson RJ, et al., Neurosci Biobehav Rev. 2021). We did all social behavioral tests during their dark cycle as many studies (Hu RK, et al., Nat Neurosci. 2021; Murugan M, et al. Cell. 2017; Karigo T, et al. Nature. 2021; Lee E, et al. J Neurosci. 2016), so the mice were housed under a light-dark cycle with light on from 20:00 to 8:00. We have rephrased it in Methods (Page 15, Line 313-315).

12) Methods, page 15, line 305: Sigma P0884 is (+/-)-propranolol, not (+)-propranolol.

We have corrected this information (Page 16, Line 328).

13) Methods, page 16: only some of the viral vectors used are listed here, while all should be listed.

We have added all viral vectors (Page 17, Line 341-347).

14) Fig 5, panel h: AAV-CAG-Cre is missing from the vector list above the brain section outline.

Thank you for the reminder. AAV-CAG-Cre has been added in Fig. 5h.

15) Supplemental figures: For the open field data, instead of "distance in the central area", it would be more informative to present % distance in the central area since the total distance of each animal varies. Also, the open field and the 0 maze present data in a manner such that quantities decrease with increased anxiety. It would be nice to be consistent with this and present the light-dark box data as time in the light box (rather than the black box).

Thank you for the suggestions. The percentage of distance in the central and total arena has been calculated and duration in the light box has been presented (Supplementary Fig. 3d, h, j, n and Supplementary Fig. 6c, g).

Reviewers' comments:

Reviewer #1 (Remarks to the Author):

The authors have adequately addressed my concerns. I support the publication of the manuscript.

Reviewer #3 (Remarks to the Author):

The authors of the manuscript by Cheng et al. have addressed most of the comments by this reviewer. However, one comment was insufficiently addressed, and several other items became apparent during the second review. These are listed below.

1) [Last part of original comment #9] "AAV-hSyn-FlpO is not selective for adrenergic neurons, so transfected neurons are likely adrenergic and non-adrenergic." The hSyn promoter is pan-neuronal rather than adrenergic neuron-specific. The authors need to acknowledge this in Results and Discussion by stating that beta-arrestin2 will also be deleted from mPFC cells innervated by non-adrenergic neurons in the vicinity of the LC that also project to the mPFC.

2) Introduction, line 54: What do the authors mean when they state "Protein synthesis in the mPFC is required for neuronal activation..."? The reference cited only indicates that it's required for SRM consolidation.

3) Introduction, line 60: the word "antagonists" is needed and grammar should be improved, perhaps by writing "which is blocked by beta-adrenergic receptor (beta-AR) antagonists".

4) Results, line 85: the authors still do not distinguish between light activation of halorhodopsin and its physiological effect. The sentence should begin with something like "Optogenetic inhibition of the eNpHR3.0+ NE terminal...".

5) Results, line 131: panel D in Fig. S3 reports distance, not time as indicated by the authors in line 131.

6) Results, line 222: the word "order" should be replaced with "older".

7) Discussion, first paragraph: there's a discrepancy in the authors' findings that needs to be added to the discussion. Given that stimulating beta-arrestin signaling rescues SRM consolidation with beta1-AR deletion but not with beta2-AR deletion, why does stimulating beta-arrestin signaling rescue SRM consolidation when mPFC NE release is inhibited, which should result in the reduction of both beta1-AR and beta2-AR signaling, with the latter not being rescued by stimulating beta-arrestin signaling?

8) Methods, line 396: the word "walls" should be added after "Plexiglas".

9) Methods, line 419: the word "for" should be deleted.

Reviewer #4 (Remarks to the Author):

The manuscript by Cheng et al describes a novel role of norepinephrine, beta-receptors and b-arrestin2 in social memory consolidation. The manuscript has been greatly improved after the first round of comments and is now well-written and can be easily followed. The authors use a combination of tools such as optogenetics, chemogenetics, pharmacology and behavior to test the role of NE, beta-receptors and Barr2 in social recognition memory and consolidation. The experiments are very

thorough since the authors use these different techniques to probe the same pathway. Overall the data are clear and suggest that NE, beta-receptors and arrestin plays an important role in SRM. I have one major concern and a few minor ones.

Major:

please mention the serotypes of the AAVs in the results and not just in the methods. For eg. AAV9-mCaMKII.....so on. This is important because it brings to light a potential caveat. one potential caveat is that similar to AAV1 scAAV-flpO used in figure 4 as an anterograde trans synaptic tool, AAV9 can also propagate transneuronally in an anterograde fashion. Thus, when AAV9-mCaMKII-iCre is used it could possibly lead to AAV9 being expressed in cell bodies of projection targets of mPFC glutamatergic neurons (striatum, thalamus etc). The authors need to show that they achieve deletion of B1AR or B2AR or ARRB2 in only mPFC and not in other brain regions i.e projection targets of mPFC.

Minor:

1. stereotaxic coordinates for viral injections are not mentioned in methods.
2. please describe here how you detected NE release. also there is no citation for the NE sensor or where it was obtained from.
3. what does 'E' and 'N' stand for in Figure S1 e and f. please describe abbreviation for empty or novel at least the first time.
4. In the methods section it says chrimson R whereas in the figures it says ChR2. those are two different opsins requiring different wavelength of light.
5. again F and N not defined in figure legend
6. it is not clear if the novel and familiar mice used in the SRM test 1 and 2 are the same or not.
7. not clear what is meant by homecage+laser. there is no mention of homecage anywhere in the manuscript.
8. nomenclature used is not accurate. half deletion is achieved by heterozygote floxed mice and should be denoted as fl/+ not fl/-
9. please briefly explain about transsynaptic labeling using AAV1, which allows knockout of Barr2 in mPFC neurons innervated by LC projections.
10. dose of CNO in N2a cells not mentioned in figure legends or results section.
11. whats GqM in the figure 5b? please use consistent labels.

Point-by-point response to Reviewers:

Reviewer #3 (Remarks to the Author):

The authors of the manuscript by Cheng et al. have addressed most of the comments by this reviewer. However, one comment was insufficiently addressed, and several other items became apparent during the second review. These are listed below.

1) [Last part of original comment #9] “AAV-hSyn-FlpO is not selective for adrenergic neurons, so transfected neurons are likely adrenergic and non-adrenergic.” The hSyn promoter is pan-neuronal rather than adrenergic neuron-specific. The authors need to acknowledge this in Results and Discussion by stating that beta-arrestin2 will also be deleted from mPFC cells innervated by non-adrenergic neurons in the vicinity of the LC that also project to the mPFC.

We thank the reviewer for the important suggestion. We have added related description in the results (Page 9, Line 190-191)

2) Introduction, line 54: What do the authors mean when they state “Protein synthesis in the mPFC is required for neuronal activation...”? The reference cited only indicates that it’s required for SRM consolidation.

We have corrected the description (Page 3, Line 57).

3) Introduction, line 60: the word “antagonists” is needed and grammar should be improved, perhaps by writing “which is blocked by beta-adrenergic receptor (beta-AR) antagonists”.

We have corrected the description (Page 4, Line 63).

4) Results, line 85: the authors still do not distinguish between light activation of halorhodopsin and its physiological effect. The sentence should begin with something like “Optogenetic inhibition of the eNpHR3.0+ NE terminal...”.

We have corrected the description (Page 5, Line 88).

5) Results, line 131: panel D in Fig. S3 reports distance, not time as indicated by the authors in line 131.

We have corrected the description (Page 7, Line 134).

6) Results, line 222: the word “order” should be replaced with “older”.

We have corrected the word (Page 11, Line 232).

7) Discussion, first paragraph: there’s a discrepancy in the authors’ findings that needs to be added to the discussion. Given that stimulating beta-arrestin signaling rescues SRM consolidation with beta1-AR deletion but not with beta2-AR deletion, why does stimulating beta-arrestin signaling rescue SRM consolidation when mPFC NE release is inhibited, which should result in the reduction of both beta1-AR and beta2-AR signaling, with the latter not being rescued by stimulating beta-arrestin signaling?

We apologize for the lack of clarity of Fig. 5g-j. We only tested the rescue effects of activation of β -arrestin signaling on SRM consolidation with inhibition of LC-mPFC NE projection or *Adrb1* deletion in the mPFC. We did not do this experiment on β 2-AR conditional knockout mice (*Adrb2^{fl/fl}*). We propose that activation of β -arrestin signaling can also rescue the impairment of SRM consolidation by *Adrb2* deletion in the mPFC.

8) Methods, line 396: the word “walls” should be added after “Plexiglas”.
We have added “walls” as suggested (Page 19, Line 399).

9) Methods, line 419: the word “for” should be deleted.
We have deleted the word “for” (Page 20, Line 424).

Reviewer #4 (Remarks to the Author):

The manuscript by Cheng et al describes a novel role of norepinephrine, beta-receptors and b-arrestin2 in social memory consolidation. The manuscript has been greatly improved after the first round of comments and is now well-written and can be easily followed. The authors use a combination of tools such as optogenetics, chemogenetics, pharmacology and behavior to test the role of NE, beta-receptors and Barr2 in social recognition memory and consolidation. The experiments are very thorough since the authors use these different techniques to probe the same pathway. Overall the data are clear and suggest that NE, beta-receptors and arrestin plays an important role in SRM. I have one major concern and a few minor ones.

Major:

please mention the serotypes of the AAVs in the results and not just in the methods. For eg. AAV9-mCaMKII.....so on. This is important because it brings to light a potential caveat.

one potential caveat is that similar to AAV1 scAAV-flpO used in figure 4 as an anterograde trans synaptic tool, AAV9 can also propagate transneuronally in an anterograde fashion. Thus, when AAV9- mCaMKII-iCre is used it could possibly lead to AAV9 being expressed in cell bodies of projection targets of mPFC glutamatergic neurons (striatum, thalamus etc). The authors need to show that they achieve deletion of B1AR or B2AR or ARRB2 in only mPFC and not in other brain regions i.e projection targets of mPFC.

We thank the reviewer for the important suggestion.

As suggested, we have added the serotypes of the AAVs in the results and figure legends.

For anterograde infection, high titer of *scAAV1-hSyn-FlpO* (1×10^{13} vector genome (vg) ml^{-1}) was applied. For local infection, regular titer ($2.0\text{-}2.5 \times 10^{12}$ vector genome (vg) ml^{-1}) of AAV9 was used. We can hardly find anterograde infection of AAV9 with regular titer. We have added the information in the methods (Page 17, Line 363-364). With injection of *AAV9-mCaMKII α -eGFP-P2A-iCre* in the mPFC, we haven't found eGFP in other brain area, such as nucleus accumbens (Fig. 2f).

Minor:

1. stereotaxic coordinates for viral injections are not mentioned in methods.

The information about stereotaxic coordinates for viral injections in the mPFC and LC was listed in **Stereotaxic surgery of Methods** (Page 17-18, Line 364-366).

2. please describe here how you detected NE release. also there is no citation for the NE sensor or where it was obtained from.

As suggested, we have added information of fiber photometry recording of NE sensor and the citations (Page 22, Line 454-460).

3. what does 'E' and 'N' stand for in Figure S1 e and f. please describe abbreviation for empty or novel at least the first time.

As suggested, we have added the definition of 'E' and 'N' in all figure legends.

4. In the methods section it says chromson R whereas in the figures it says ChR2. those are two different opsins requiring different wavelength of light.

We apologize for the lack of clarity of method section about optogenetic stimulation. We have reorganized this part (Page 21-22).

ChrimsonR and ChR2 are two different channelrhodopsins. ChR2 reacts rapidly to brief pulses of blue light (473 nm), and Chrimson responds to yellow light (590 nm). Stimulation of these channelrhodopsins can induce depolarizing photocurrents to mediate action potentials. We used ChR2 in all behavioral tests. We used ChrimsonR in the experiment of photometry recording of NE release to avoid the influence of optical stimulation when we simultaneously recorded NE2h transient (473 nm).

5. again F and N not defined in figure legend

As suggested, we have added the definition of 'F' and 'N' in all figure legends.

6. it is not clear if the novel and familiar mice used in the SRM test 1 and 2 are the same or not.

The familiar mouse is the same used in sociability test and novel mouse is different in each SRM test. We have added this information in the method (Page 20, Line 420-421).

7. not clear what is meant by homecage+laser. there is no mention of homecage anywhere in the manuscript.

In western blotting experiment, the mice stayed at homecage with or without optical stimulation were used as the control for sociability tests. We have added this information in the method (Page 25, Line 537-538).

8. nomenclature used is not accurate. half deletion is achieved by heterozygote floxed mice and should be denoted as fl/+ not fl/-

Thank you for the correction. We have replaced fl/- with fl/+ in manuscript and figures.

9. please briefly explain about transsynaptic labeling using AAV1, which allows knockout of Barr2 in mPFC neurons innervated by LC projections.

As suggested, we have added information about transsynaptic labeling using AAV1 in the results (Page 9, Line 183-187).

10. dose of CNO in N2a cells not mentioned in figure legends or results section.

We have added the dose of CNO in culture experiments in results section and figure legends.

11. whats GqM in the figure 5b? please use consistent labels.

Thank you for the suggestion. We have replaced GqM with rM3Darr in Fig. 5b.

REVIEWERS' COMMENTS:

Reviewer #3 (Remarks to the Author):

The revisions of the manuscript now make it appropriate for publication

Reviewer #4 (Remarks to the Author):

the authors have addressed all my concerns. please make sure to check for spelling errors.